

# PhoStream: Benchmarking Real-World Streaming for Omnimodal Assistants in Mobile Scenarios

Xudong Lu [1]   Huankang Guan [2]   Yang Bo [2]   Jinpeng Chen [2]   Xintong Guo [2]   Shuhan Li [2]   Fang Liu [3]
Peiwen Sun [1]   Xueying Li [4]   Wei Zhang [4]   Xue Yang [4]   Rui Liu [2]   Hongsheng Li [1]

## Abstract

Multimodal Large Language Models excel at offline audio-visual understanding, but their ability to serve as mobile assistants in continuous real-world streams remains underexplored. In daily phone use, mobile assistants must track streaming audio-visual inputs and respond at the right time, yet existing benchmarks are often restricted to multiple-choice questions or use shorter videos. In this paper, we introduce **PhoStream**, the first mobile-centric streaming benchmark that unifies on-screen and off-screen scenarios to evaluate video, audio, and temporal reasoning. PhoStream contains 5,572 open-ended QA pairs from 578 videos across 4 scenarios and 10 capabilities. We build it with an Automated Generative Pipeline backed by rigorous human verification, and evaluate models using a realistic Online Inference Pipeline and LLM-as-a-Judge evaluation for open-ended responses. Experiments reveal a temporal asymmetry in LLM-judged scores (0–100): models perform well on Instant and Backward tasks (Gemini 3 Pro exceeds 80), but drop sharply on Forward tasks (16.40), largely due to early responses before the required visual and audio cues appear. This highlights a fundamental limitation: current MLLMs struggle to decide *when* to speak, not just *what* to say. Code and datasets will be made publicly accessible at https://github.com/Lucky-Lance/PhoStream.

## 1. Introduction

Recent advances in Multimodal Large Language Models (MLLMs) have achieved strong performance on visual un-

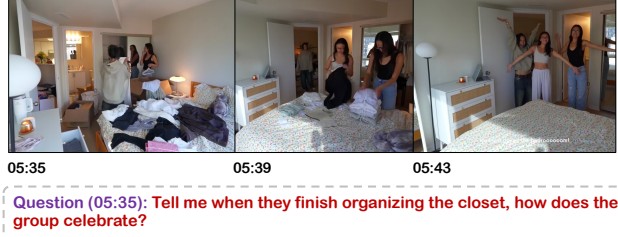

*Figure 1.* **Motivating example from PhoStream.** The question is asked at 05:35, but the key evidence appears later. Gemini 3 Pro answers immediately and is wrong, while Doubao-Seed-1.6 keeps watching and answers correctly at 05:43.

derstanding tasks. Representative commercial models include Gemini 3 Pro (Google DeepMind, 2025) and GPT-4o (Hurst et al., 2024), while open-source models have also demonstrated competitive capabilities, such as Qwen3-VL (Bai et al., 2025), MiniCPM-V 4.5 (Yu et al., 2025) and InternVL 3.5 (Wang et al., 2025a). However, most existing MLLMs are primarily designed for static images or offline videos. To better support real-world online video streams, streaming systems such as VideoLLM-online (Chen et al., 2024), Dispider (Qian et al., 2025), MMDuet2 (Wang et al., 2025b), and StreamingVLM (Xu et al., 2025c) have been proposed. These streaming systems can respond to the current visual state in real time, and some further enhance this capability by recalling past context or deferring responses until sufficient future evidence becomes available.

As devices people use every day, smartphones are among the most effective and accessible platforms for deploying streaming capabilities to support everyday needs. Mobile AI assistants like Doubao (ByteDance, 2026) exemplify this setting, where users rely on their phones to follow in-app tutorials or navigate outdoors. To enable these interactions, assistants must process multimodal inputs by integrating visual and audio streams, as demonstrated by omnimodal models such as Gemini 3 Pro (Google DeepMind, 2025) and Qwen3-Omni (Xu et al., 2025b). More critically, they must exercise temporal reasoning across contexts to decide when and how to respond. This requirement for dynamic temporal

[1]CUHK MMLab [2]Huawei Research [3]City University of Hong Kong [4]Shanghai Jiao Tong University. Correspondence to: Rui Liu <liu.rui2@huawei.com>, Hongsheng Li <hsli@ee.cuhk.edu.hk>.

*Proceedings of the 43rd International Conference on Machine Learning*, Seoul, South Korea. PMLR 306, 2026. Copyright 2026 by the author(s).

*Table 1.* **Comparison of video understanding benchmarks for streaming scenarios.** PhoStream is the only benchmark that targets mobile-centric scenarios while covering the full temporal scope with open-ended QA. Notably, it features the longest average video duration and the highest question density, suggesting potentially stronger inter-question dependence within each video.

| Benchmark | Modality | #Videos | Avg. Dur. (min) | #Ques. | Avg. Q/V | Mobile Centric | Open Ended | Temporal Scope | | |
| --- | --- | --- | --- | --- | --- | --- | --- | --- | --- | --- |
| | | | | | | | | Backward | Instant | Forward |
| StreamingBench | V, A | 900 | 9.7 | 4,500 | 5.0 | ✗ | ✗ | ✓ | ✓ | ✓ |
| OVO-Bench | V | 644 | 7.9 | 2,814 | 4.4 | ✗ | ✗ | ✓ | ✓ | ✓ |
| OmniMMI | V, A | 1,121 | 5.4 | 2,290 | 2.0 | ✗ | ✓* | ✓ | ✓ | ✓ |
| ProactiveVideoQA | V, A | 1,377 | 2.1 | 1,427 | 1.0 | ✗ | ✓ | ✗ | ✗ | ✓ |
| **PhoStream (Ours)** | **V, A** | 578 | **13.3** | **5,572** | 9.6 | ✓ | ✓ | ✓ | ✓ | ✓ |

*: Only for specific sub-tasks. **V**: Video, **A**: Audio.

*Table 2.* **Statistics of the evaluation samples.** We report the sample counts for Instant, Backward, and Forward tasks across four mobile-centric scenarios. Notably, the Forward task comprises the largest portion of the benchmark (approx. 50%).

| Scenario | Temporal Scope | | | Total |
| --- | --- | --- | --- | --- |
| | Instant | Backward | Forward | |
| YouTube Vlog | 997 | 629 | 1,631 | **3,257** |
| Phone Tutorial | 417 | 255 | 673 | **1,345** |
| Phone Record | 150 | 128 | 296 | **574** |
| EgoBlind | 83 | 111 | 202 | **396** |
| **Total** | **1,647** | **1,123** | **2,802** | **5,572** |

awareness fundamentally distinguishes mobile streaming assistance from traditional offline video question answering.

To systematically evaluate streaming capabilities, several benchmarks have been proposed in recent studies. However, as shown in Tab. 1, existing benchmarks fail to capture the complexity of real-world mobile assistant scenarios adequately. In daily human-phone interactions, users pose diverse open-ended questions, yet StreamingBench (Lin et al., 2024) and OVO-Bench (Niu et al., 2025) rely on multiple-choice formats that fail to capture this natural interaction pattern. While OmniMMI (Wang et al., 2025d) includes some open-ended questions, it constrains them to predefined task types such as state grounding and speaker identification, which do not fully align with the diverse real-world scenarios encountered in everyday mobile usage. ProactiveVideoQA (Wang et al., 2025c) focuses exclusively on proactive interactions, and most of its egocentric videos lack the audio modality. More importantly, existing benchmarks lack full coverage of the two complementary categories that constitute the complete mobile experience: on-screen and off-screen interactions. On-screen scenarios involve device tutorials and app operation flows captured through screen recordings, where users need guidance on interface navigation and feature usage. Off-screen scenarios include everyday videos captured by phone cameras, such as daily vlogs, activities, and entertainment content that users typically watch and interact with. To address these gaps, we introduce PhoStream, the first mobile-centric benchmark unifying on-screen and off-screen scenarios to rigorously evaluate video, audio, and temporal reasoning capabilities.

PhoStream is a mobile-centric streaming benchmark with 5,572 open-ended QA pairs from 578 videos across 4 sce-

narios, covering 10 capabilities from perception to reasoning. Compared with other open-ended benchmarks such as ProactiveVideoQA (Wang et al., 2025c), which averages 2.1 minutes per clip, PhoStream videos average 13.3 minutes, requiring models to maintain context over longer time spans rather than relying on short snippets. Moving beyond multiple-choice questions, we introduce an LLM-as-a-Judge evaluation framework that rewards natural, well-timed, and precise assistant-style responses.

PhoStream is built on a carefully designed Automated Generative Pipeline with rigorous human verification. Gemini 3 Pro analyzes the stream to generate candidate questions and timestamps and performs an initial self-verification, after which 10 human experts conduct two rounds of review to correct errors and finalize the annotations. We evaluate models with an Online Inference Pipeline that updates the video stream every 1 second, and issues each query only once at its questioning timestamp. The model may answer immediately or keep watching and decide when to respond. This enables a unified protocol for Backward, Instant, and Forward questions, where Early Response and No Response cases are assigned a score of 0. Our evaluation exposes a temporal asymmetry in current MLLMs: Gemini 3 Pro answers prematurely in 79.12% of Forward cases, and Qwen3-Omni reaches an Early Response rate of 97.89%, indicating that while current models exhibit strong visual and audio perception capabilities, they still lack the patience and temporal reasoning needed for reliable proactive assistance. As illustrated in Fig. 1, this impatience often leads models to answer Forward questions before the necessary evidence appears. Our contributions are summarized as follows:

**1)** We introduce PhoStream, the first mobile-centric streaming benchmark that unifies on-screen and off-screen scenarios. With videos averaging 13.3 minutes spanning 4 diverse scenarios and evaluating 10 distinct capabilities, PhoStream enables rigorous assessment of multimodal understanding, including video, audio, and temporal reasoning.

**2)** We develop a scalable Automated Generative Pipeline that reduces the cost of manual annotation. Additionally, we implement a realistic Online Inference Pipeline coupled with an LLM-as-a-Judge evaluation framework to ensure rigorous assessment under streaming conditions.

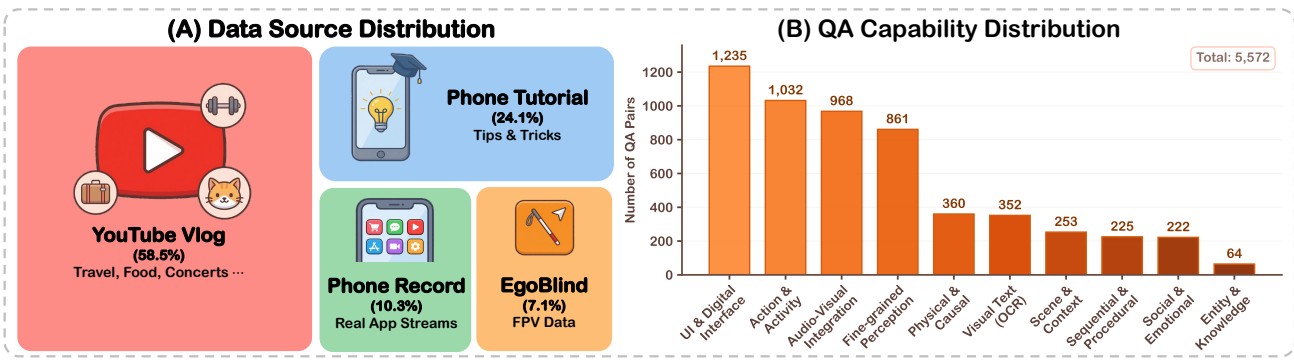

Figure 2. **Distribution of data sources and QA capability in PhoStream.** The dataset comprises 5,572 question-answer pairs from 578 videos spanning 4 mobile-centric scenarios (YouTube vlogs, phone tutorials, app recordings, and egocentric videos). It covers 10 distinct capabilities, such as UI navigation, action recognition, audio-visual integration, etc.

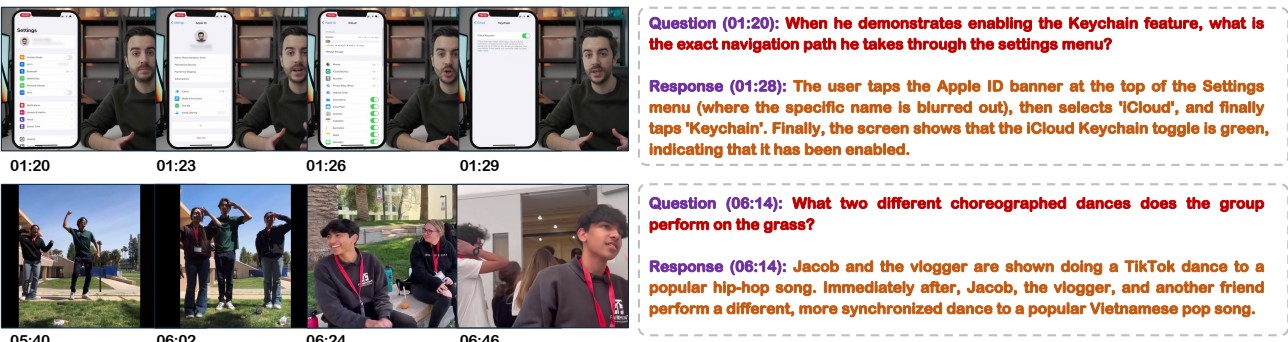

Figure 3. **Data examples in PhoStream.** (Top) An on-screen scenario involving a Forward task, where the model must wait for subsequent UI operations or specific interface states. (Bottom) An off-screen scenario featuring a Backward task, requiring the model to trace back through the video history to answer questions about past events.

**3)** We conduct comprehensive analyses and uncover a fundamental yet overlooked failure. Current models are too impatient. Instead of waiting for future events to occur, they tend to guess immediately. We identify this as Early Response bias, showing that models struggle to decide when to speak, not just what to say.

## 2. Related Works

**Multimodal Large Language Models.** Recent advancements have significantly expanded MLLM capabilities. Proprietary models such as GPT-4o (Hurst et al., 2024) and Gemini 3 Pro (Google DeepMind, 2025) achieve state-of-the-art performance in multimodal understanding, demonstrating impressive success across various complex tasks, including text recognition (Mathew et al., 2021; Fu et al., 2024), video understanding (Fu et al., 2025; Yu et al., 2019), mathematical reasoning (Wang et al., 2024; Zhang et al., 2024), audio analysis (Zhang et al., 2022; Anastassiou et al., 2024), and spatial understanding (Team et al., 2025; Wu et al., 2025). Meanwhile, open-source models such as InternVL 3.5 (Wang et al., 2025a), MiniCPM-V 4.5 (Yu et al., 2025), Qwen3-Omni (Xu et al., 2025b), and DeepSeek-VL2 (Wu et al., 2024) have also demonstrated competi-

tive capabilities in these domains. Despite these perceptual breakthroughs, most current MLLMs are restricted to offline processing of pre-recorded content and cannot perform real-time online reasoning on continuous streams.

**Streaming MLLMs and Benchmarks.** To address the limitation of offline processing, research on streaming MLLMs has emerged from both modeling and evaluation perspectives. On the modeling front, frameworks like VideoLLM-online (Chen et al., 2024) and StreamingVLM (Xu et al., 2025c) optimize memory and processing efficiency for streaming dialogues. Dispider (Qian et al., 2025) resolves perception-reaction conflicts through an asynchronous architecture, while MMDuet2 (Wang et al., 2025b) utilizes multi-turn reinforcement learning to train models to autonomously decide whether to respond or remain silent. On the evaluation front, StreamingBench (Lin et al., 2024), OVO-Bench (Niu et al., 2025), and OmniMMI (Wang et al., 2025d) evaluate streaming video understanding and proactive reasoning, while ProactiveVideoQA (Wang et al., 2025c) specifically targets evaluation that better reflects user experience in proactive interaction scenarios. However, these benchmarks primarily focus on general video understanding and lack specialized evaluation of mobile-centric streaming scenarios with open-ended model interactions.

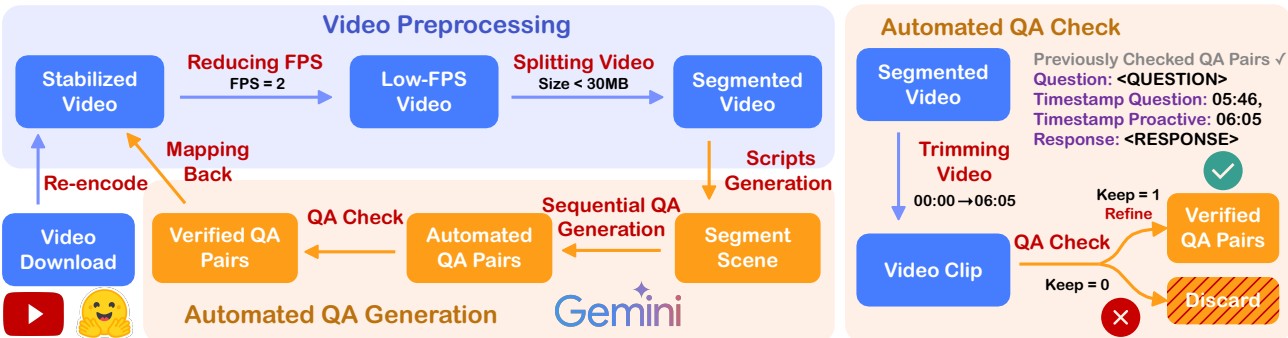

*Figure 4.* **Workflow of the PhoStream Automated Generative Pipeline.** The process integrates three primary stages: video preprocessing for stabilization and segmentation, automated QA generation, and a multi-step verification process. Gemini 3 Pro is utilized as the core engine for both initial data annotation and subsequent quality checks. This approach ensures high-quality data by using precise temporal verification to refine or discard candidate QA pairs, effectively converting raw mobile video streams into verified benchmarks.

**MLLMs for Mobile Scenarios.** Current mobile research is evolving along three key directions to enhance user experience and system efficiency. First, online mobile assistants like Doubao (ByteDance, 2025; 2026) and Gemini (Google DeepMind, 2025) enable real-time multimodal interaction and complex task handling. Second, on-device MLLMs such as MiniCPM-V 4.5 (Yu et al., 2025), OpenELM (Mehta et al., 2024), and BlueLM-V-3B (Lu et al., 2025a) optimize for edge deployment to reduce latency and improve privacy. Third, mobile GUI agents such as Mobile-Agent (Ye et al., 2025), AMEX (Chai et al., 2024), and Android-World (Rawles et al., 2024) focus on autonomous UI manipulation and multi-step task execution. PhoStream extends the first research direction by providing a streaming benchmark that evaluates temporal reasoning and multimodal understanding, emphasizing when and how models should respond under evolving contexts.

## 3. The PhoStream Benchmark

In this section, we present a comprehensive breakdown of the PhoStream benchmark, including data collection (Sec. 3.1), annotation methodology (Sec. 3.2), and statistical analysis (Sec. 3.3). We also detail our streaming inference framework (Sec. 3.4) and evaluation protocol (Sec. 3.5).

### 3.1. Data Collection and Sources

PhoStream consists of both on-screen and off-screen videos with four representative categories for real-world mobile streaming scenarios: YouTube Vlog, Phone Tutorial, Phone Record, and EgoBlind, as shown in Fig. 2. For the first two categories, we carefully curate YouTube videos covering diverse daily-life content (e.g., daily vlogs, live concerts, workout routines, travel) and mobile device operations (e.g., iPhone guides, app tutorials), prioritizing popular and high-quality videos. To evaluate models' understanding of visual and audio content rather than text, we exclude external subtitle files. Phone Record contains app usage videos that

we record by ourselves, while EgoBlind incorporates first-person videos from the EgoBlind dataset (Xiao et al., 2025), collected from video-sharing platforms such as BiliBili and TikTok, featuring blind users' perspectives. After careful curation and annotation across all categories, we sample and construct the final benchmark.

### 3.2. Task Definition and Annotation Pipeline

In this subsection, we define online video QA with strict timestamp cutoffs for realistic streaming evaluation. We introduce an automated pipeline to generate and verify timestamped QA pairs from long videos using state-of-the-art MLLMs. As a final quality check, 10 human experts conduct a two-round review, removing unsuitable videos and revising or deleting problematic QA pairs.

**Task Definition**. We study online video understanding, where a model must answer based on the visual and audio evidence available up to a time boundary, rather than assuming access to the full video. We group questions into Backward, Instant, and Forward tasks. Backward questions use only past context and include two types: retrospective questions that ask about a few specific earlier events within a limited prior context, and comprehensive questions that integrate evidence across a longer span from the beginning of the video up to the question timestamp. Instant questions focus on the current 1–2 second window and can be answered immediately after the question is posed. Forward questions are asked before the needed evidence appears, so the model should wait and respond only when the question becomes answerable. To prevent future information leakage, we verify each QA pair using a cutoff timestamp. For Backward and Instant questions, the cutoff is the question timestamp (Timestamp Question). For Forward questions, the cutoff is the earliest time when the question can be answered (Timestamp Proactive). We keep a sample only if the answer is fully supported within the cutoff timestamp.

**Automated Generative Pipeline**. To reduce large-scale

manual annotation, we develop an automated pipeline that uses Gemini 3 Pro to generate timestamped, verifiable QA pairs for long-form videos. Fig. 4 illustrates the overall workflow. After downloading each video, we re-encode it to MP4 with HEVC (H.265) using NVIDIA NVENC to stabilize frame timing, align timestamps, and mitigate common corruption issues. We then resample the video to 2 FPS and use this version as the canonical stream for all subsequent generation steps, increasing the sampling density relative to Gemini's default 1 FPS and reducing misses on fast actions. We further split the 2 FPS MP4 into segments smaller than 30MB using MP4Box (GPAC Team, 2024), enabling reliable parallel processing under storage and transfer constraints.

For each segment, Gemini 3 Pro generates scene summaries and coarse step-wise scripts. Based on the scripts and the corresponding video segment, we then generate candidate QA pairs using a prompting template with question examples, covering (but not limited to) UI navigation, actions & activities, audio-video integration, and fine-grained visual perception. Finally, an automated checker verifies each QA pair by re-evaluating it using only the video content up to a cutoff timestamp, together with the preceding dialogue history. For Backward and Instant questions, the cutoff is the question timestamp (Timestamp Question). For Forward questions, the cutoff is the earliest timestamp at which the question can be answered (Timestamp Proactive). Under this cutoff, the checker accepts the QA if it remains correct, revises it when the required evidence is still available, and discards it otherwise. This prevents future information leakage and ensures that each answer is supported within the specified time cutoff. We retain only verified samples, link them back to their source video segments, and store complete metadata. Example QA pairs are shown in Fig. 3.

**Human Verification**. To ensure the correctness and safety of the benchmark, we perform human verification on top of the automated generative pipeline. We start from long high-FPS videos and downsample them to 2 FPS for efficient processing, then split the downsampled videos into segments of at most 30 MB. During human verification, we map each sample timestamp from the 2 FPS segments back to the original stabilized high-FPS video and extract a short high-FPS clip around the mapped time point. Each candidate sample includes a question, an answer, a task type label (Backward, Instant, or Forward), the relevant timestamps (Timestamp Question and Timestamp Proactive, if any), the linked source segment, the extracted high-FPS clip, and the dialogue history used for generation, with a link to the full original video for additional context when needed. We recruit 10 human experts with experience in multimodal video understanding to run two rounds of review. In each round, experts check whether the question is clear and can be answered using only the visual and audio

evidence within the allowed time window, and whether the answer is correct and complete. For Backward and Instant questions, they verify that the answer is supported by the content up to Timestamp Question. For Forward questions, they verify that Timestamp Proactive is the earliest time when the question becomes answerable and that earlier content does not already reveal the answer. When problems are found, experts revise the sample by editing the question or answer, fixing the task type label, and adjusting timestamps to remove future information leakage. If a sample is unclear, has more than one reasonable interpretation, lacks enough evidence, or the experts cannot reach agreement, we discard it. We also remove any samples or videos that contain pornography, explicit sexual content, excessive violence, political propaganda, hate, harassment, or other unsafe content to support responsible release. After two rounds of verification and filtering, we keep only samples that are accurate, clear, and fully supported by the video and audio evidence within the specified time boundary.

### 3.3. Dataset Statistics and Analysis

In this subsection, we report detailed statistics of the automated pipeline and two-round human verification. We then summarize the final dataset and coverage of PhoStream.

**Automated Pipeline and Human Verification**. We build PhoStream through automated generation followed by careful human verification. Starting from 523 unique source videos, we split them into 589 segment files and use Gemini 3 Pro to generate 6,042 timestamped QA candidates. Our automatic QA check removes 218 candidates that violate the streaming constraint or fail basic quality checks, including answers inconsistent with the video or audio evidence, hallucinated details, or reliance on content after the cutoff timestamp. This step enforces the streaming rule and reduces future leakage, leaving 5,824 candidates for human review. We recruit 10 human experts and conduct two rounds of review to remove unsuitable videos and improve QA clarity, correctness, and timestamp accuracy. After the first round, we retain 578 videos and 5,611 QA pairs. After the second round, we finalize 5,572 QA pairs.

**Final Dataset and Coverage**. The final benchmark contains 5,572 open-ended QA pairs from 578 videos across four scenarios, including YouTube Vlog, Phone Tutorial, Phone Record, and EgoBlind. The average video length is 13.3 minutes. This setting is more challenging than existing short-clip benchmarks, as models cannot rely on brief segments and must instead track events over extended streams with limited memory. Fig. 2(A) summarizes the QA distribution across scenarios. The benchmark includes both on-screen and off-screen scenarios. On-screen scenarios cover two types of content. One type is device tutorial videos curated from online sources. The other type is app

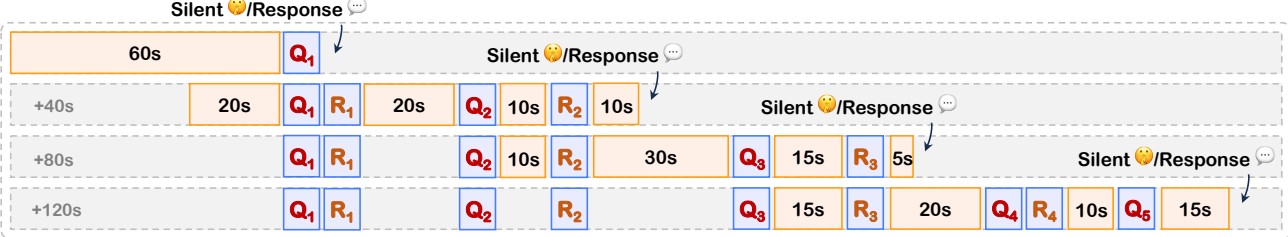

*Figure 5.* **Online Inference Pipeline in PhoStream**. The model processes continuous video streams sequentially in 1-second intervals while maintaining a sliding memory window (e.g., 60s) to manage context. Unlike traditional approaches (Lin et al., 2024; Niu et al., 2025) that repetitively query the model at every time step, we issue the query only once at the relevant questioning timestamp, allowing the model to autonomously determine the response timing, thereby enabling a unified strategy to handle diverse temporal logics.

operation workflows that we record through screen capture. Off-screen scenarios are videos recorded with smartphone cameras and collected from online sources. They include vlogs, travel videos, and other videos that capture everyday life. To support finer-grained analysis beyond scenario labels, we also annotate each QA with a capability category by Qwen3-235B-A22B (Yang et al., 2025). Fig. 2(B) reports the capability distribution, which helps diagnose failure modes. For example, a model may perform well on general actions but struggle when the answer requires reading on-screen text or linking audio to a visual event.

Tab. 2 reports the number of Backward, Instant, and Forward questions in each scenario. Forward questions account for the largest share, close to half of the dataset. This design emphasizes proactive streaming behavior, where the model must produce correct answers while also avoiding responses before the supporting evidence appears. Combined with open-ended answers, our benchmark better reflects real assistant interactions than multiple-choice streaming benchmarks such as StreamingBench and OVO-Bench, and it avoids cues from answer options. Additional statistics, including a word cloud and the question time distribution, are provided in Fig. A.1 in the Appendix.

### 3.4. Online Inference Pipeline

We design an Online Inference Pipeline that matches real mobile use, as shown in Fig. 5. The model consumes video and audio strictly in time order, with the stream updated every second. At any time, the model can access only the most recent 60 seconds of video via a sliding window to reduce memory usage, while the full text dialogue history (i.e., all past questions and model responses) is always available. For each question, we send the query to the model only once at its questioning timestamp. After the query is issued, the model continues to watch the stream with the same 1-second updates. At each update, it decides to stay silent (strictly output the exact word "Silent", as required by the system prompt) or to answer the issued query. If the accessible context is already sufficient at the questioning timestamp, the model should answer immediately. On the other hand,

if the evidence needed for the answer has not appeared yet, the model should stay silent and keep monitoring until later evidence becomes visible. This setup supports Backward, Instant, and Forward questions with a single procedure.

### 3.5. Evaluation Protocol

We evaluate streaming performance by enforcing strict timestamp-based response windows and scoring valid outputs with an LLM-as-a-Judge rubric.

**Response Window**. For streaming evaluation, we define a response window for each QA. For Backward and Instant questions, a response is valid only if it occurs exactly at Timestamp Question. For Forward questions, a response is valid if it occurs within a 2-second window starting at Timestamp Proactive. We treat placeholder responses[1] as non-informative and skip them during evaluation. Any response other than Silent or a placeholder occurring before the ground-truth timestamp is labeled as an Early Response (ER) with a score of 0. If the model produces no response other than Silent or a placeholder within the response window, it is labeled as No Response (NR) with a score of 0. Only non-silent, non-placeholder responses within the window are considered valid and sent for judging.

**Judging Rubric**. Open-ended questions often do not have a single definitive ground-truth answer. Consequently, many benchmarks adopt LLM-as-a-Judge for evaluation (Li et al., 2023; Zheng et al., 2023; Lu et al., 2025b; Liu et al., 2024). We follow this setting and evaluate model outputs using a fixed rubric and the judging prompt in Listing A.2. For each prediction, the judging LLM is given the question, the model output, and a reference answer, and assigns an integer score from 0 to 5. The score measures whether the output provides a reasonable explanation that is relevant to the question, factually plausible, and supported by clear causal reasoning. The rubric emphasizes the core reasoning rather than exact overlap with the reference. A score of 5 indicates a correct and complete explanation, while lower scores reflect missing

---

[1]Placeholder responses (listed in Listing A.1) are ignored during response matching and window validation.

*Table 3.* **Main evaluation results on PhoStream.** We report evaluation scores for Instant, Backward, and Forward tasks, along with the Overall average. The right section analyzes model behavior in Forward tasks: *Early Response* (ER, ↓), *No Response* (NR, ↓), and *Partly Correct* (PC, ↑). The best result in each category is **bolded**.

| Model | Param | Evaluation Score (↑) | | | Overall (↑) | Forward Task Analysis (%) | | |
|---|---|---|---|---|---|---|---|---|
| | | Instant | Backward | Forward | | ER (↓) | NR (↓) | PC (↑) |
| *Proprietary Multimodal Models* | | | | | | | | |
| Gemini 3 Pro (Google DeepMind, 2025) | - | **80.83** | **82.19** | 16.40 | 48.70 | 79.12 | **0.11** | 20.77 |
| Doubao-Seed-1.6 (ByteDance, 2025) | - | 71.28 | 62.94 | **44.26** | 56.01 | **29.76** | 13.38 | **56.85** |
| Doubao-Seed-1.8 (ByteDance, 2026) | - | 80.45 | 77.31 | 33.38 | **56.15** | 56.46 | 2.36 | 41.18 |
| *Open-source Multimodal Models* | | | | | | | | |
| Qwen2.5-Omni-7B (Xu et al., 2025a) | 7B | 67.71 | 65.20 | 1.81 | 34.06 | **42.65** | 43.50 | **13.85** |
| Qwen3-VL-8B (Bai et al., 2025) | 8B | 75.22 | 71.50 | **7.18** | **40.25** | 85.44 | 3.50 | 11.06 |
| Qwen3-VL-30B-A3B (Bai et al., 2025) | 30B | 73.38 | 69.46 | 5.25 | 38.33 | 91.33 | 1.18 | 7.49 |
| Qwen3-Omni-30B-A3B (Xu et al., 2025b) | 30B | **77.18** | **77.24** | 1.26 | 39.02 | 97.89 | **0.07** | 2.03 |
| *Open-source Multimodal Streaming Models* | | | | | | | | |
| Dispider (Qian et al., 2025) | 7B | **44.24** | **42.90** | **3.53** | **23.50** | 68.52 | 21.20 | 10.28 |
| VideoLLM-online-8B (Chen et al., 2024) | 8B | 24.88 | 24.72 | 0.00 | 12.34 | 99.54 | **0.43** | 0.04 |
| MMDuet2 (Wang et al., 2025b) | 3B | 8.76 | 8.30 | 1.39 | 4.96 | **28.55** | 59.21 | **12.24** |

causal links, insufficient support, factual errors, or off-topic responses. For comparability with benchmarks reported on a 0–100 scale, we multiply the 0–5 score by 20 and report the average. We compute the final scores for Instant, Backward, and Forward tasks using the same procedure.

## 4. Experiment

In this section, we present a series of experiments on PhoStream. We describe the experimental setup (Sec. 4.1), report baseline results (Sec. 4.2), conduct an audio ablation study (Sec. 4.3), and perform a human test to validate the LLM-as-a-Judge evaluation (Sec. 4.4). More fine-grained results are provided in the Appendix, with breakdowns by scenario subsets and capability categories (Sec. A.3), as well as failure cases and qualitative examples (Sec. A.6).

### 4.1. Experiment Setup

We evaluate three categories of baseline models with the Online Inference Pipeline and LLM-as-a-Judge evaluation, including proprietary MLLMs, open-source MLLMs, and open-source streaming MLLMs[2]. We leverage Qwen3-235B-A22B-Instruct (Yang et al., 2025) as the judging LLM and report Instant, Backward, Forward, and Overall scores for all models. For further analysis of the Forward setting, we also report the proportions of Early Response (ER, ↓), No Response (NR, ↓), and Partly Correct (PC, ↑). ER denotes any response other than Silent or a placeholder that occurs before Timestamp Proactive. NR denotes that the model produces no response other than Silent or a placeholder within the response window. The remaining cases where the model responds within the valid window are categorized as Partly Correct (PC). In practice, to improve efficiency, for

[2]We omit StreamingVLM (Xu et al., 2025c) from our experiments since it does not support open-ended question answering.

Instant and Backward questions we construct the context only at Timestamp Question and run inference once. For Forward questions, we use a 2-second response window and run inference at 6 time points, Timestamp Question and Timestamp Proactive, each with offsets of 0, +1, and +2 seconds.

### 4.2. Main Results

Tab. 3 reports baseline results on the full PhoStream benchmark. Proprietary models achieve higher Overall scores than open-source models, and this advantage persists across settings. Models also perform substantially better in the Instant and Backward settings than in the Forward setting. After a detailed analysis of the results, we observe the following consistent patterns:

**1)** Forward is the bottleneck. Most models obtain relatively high scores on Instant and Backward, but their Forward scores are much lower. For example, Gemini 3 Pro achieves 80.83 and 82.19 on Instant and Backward, yet only 16.40 on Forward. Similarly, Qwen3-Omni scores 77.18 and 77.24 on Instant and Backward, but only 1.26 on Forward.

**2)** Early Response is a major driver of Forward degradation for many models. Models optimized for streaming dialogue tend to produce continuous descriptive commentary and respond prematurely with narration-like outputs (e.g., VideoLLM-online), whereas our benchmark requires assistant behavior that waits to answer until sufficient cues are available. Early Response is also severe in strong general MLLMs, such as Qwen3-Omni with 97.89% ER, Qwen3-VL-30B-A3B with 91.33%, and Gemini 3 Pro with 79.12%.

**3)** Better Instant and Backward performance often comes with stronger Early Response tendency. Models that are more capable when evidence is already available tend to answer more aggressively in Forward settings, thus increasing

*Table 4.* **Ablation study on audio modality.** We compare model performance with and without audio input across Instant, Backward, and Forward tasks. The Δ rows show the performance difference (with audio - without audio), where positive values indicate improvement for PC and evaluation score metrics, and negative values indicate improvement for ER/NR metrics.

| Model | Audio | Evaluation Score (↑) | | | Overall (↑) | Forward Task Analysis (%) | | |
| --- | --- | --- | --- | --- | --- | --- | --- | --- |
| | | Instant | Backward | Forward | | ER (↓) | NR (↓) | PC (↑) |
| *Gemini 3 Pro* | | | | | | | | |
| Gemini 3 Pro | ✓ | **80.83** | **82.19** | 16.40 | **48.70** | 79.12 | **0.11** | 20.77 |
| Gemini 3 Pro | ✗ | 77.46 | 72.84 | **18.79** | 47.03 | **75.34** | 0.71 | **23.95** |
| Δ | | **+3.37** | **+9.35** | -2.39 | **+1.67** | +3.78 | **-0.60** | -3.18 |
| *Qwen3-Omni-30B-A3B* | | | | | | | | |
| Qwen3-Omni-30B-A3B | ✓ | **77.18** | **77.24** | 1.26 | **39.02** | 97.89 | **0.07** | 2.03 |
| Qwen3-Omni-30B-A3B | ✗ | 74.10 | 70.92 | **1.33** | 36.87 | **97.68** | 0.14 | **2.18** |
| Δ | | **+3.08** | **+6.32** | -0.07 | **+2.15** | +0.21 | **-0.07** | -0.15 |

*Table 5.* **Human evaluation results on PhoStream (sampled dataset).** Results based on 200 sampled videos (50 per scenario) with human annotations. We report evaluation scores for Instant, Backward, and Forward tasks, along with the Overall average.

| Model | Param | Evaluation Score (↑) | | | Overall (↑) | Forward Task Analysis (%) | | |
| --- | --- | --- | --- | --- | --- | --- | --- | --- |
| | | Instant | Backward | Forward | | ER (↓) | NR (↓) | PC (↑) |
| *Proprietary Multimodal Models* | | | | | | | | |
| Gemini 3 Pro | - | 79.83 | 82.27 | 17.04 | 48.97 | 78.70 | **0.00** | 21.30 |
| Doubao-Seed-1.6 | - | 73.50 | 70.42 | **47.86** | **59.92** | **29.25** | 12.03 | **58.72** |
| Doubao-Seed-1.8 | - | **82.08** | **83.98** | 36.91 | 59.91 | 53.09 | 2.98 | 43.93 |
| *Open-source Multimodal Models* | | | | | | | | |
| Qwen2.5-Omni-7B | 7B | 70.49 | 68.24 | 2.23 | 35.77 | **46.03** | 38.30 | **15.67** |
| Qwen3-VL-8B | 8B | 76.26 | 75.05 | **7.15** | **41.36** | 84.33 | 4.64 | 11.04 |
| Qwen3-VL-30B-A3B | 30B | 74.56 | 71.39 | 5.58 | 39.26 | 90.18 | 1.99 | 7.84 |
| Qwen3-Omni-30B-A3B | 30B | **76.77** | **77.59** | 1.10 | 39.07 | 98.01 | **0.11** | 1.88 |
| *Open-source Multimodal Streaming Models* | | | | | | | | |
| Dispider | 7B | **40.98** | **39.63** | **2.72** | **21.49** | 70.42 | 19.98 | 9.60 |
| VideoLLM-online-8B | 8B | 21.36 | 22.78 | 0.00 | 11.00 | 99.23 | **0.77** | 0.00 |
| MMDuet2 | 3B | 9.51 | 9.12 | 1.21 | 5.26 | **30.68** | 55.19 | **14.13** |

the ER rate. For instance, Qwen3-Omni shows strong Instant and Backward performance but exhibits a very high ER rate of 97.89%. Compared with Doubao-Seed-1.6, Doubao-Seed-1.8 performs better on Instant and Backward but answers much earlier and more frequently, which contributes to its lower Forward score.

**4)** No Response is another distinct Forward failure mode. While many models fail due to premature answering, some instead miss the response window and produce no non-silent output, leading to high No Response rates. For example, MMDuet2 (likely constrained by its 3B scale) has 59.21% NR, and Qwen2.5-Omni-7B has 43.50% NR, suggesting a bottleneck in response triggering rather than impatience.

### 4.3. Audio Ablation Analysis

To verify the effectiveness of the audio modality, we conduct an ablation study on two representative omni models, Gemini 3 Pro and Qwen3-Omni, which natively support audio-visual inputs. Tab. 4 shows that adding audio improves Instant and Backward scores, leading to higher Overall performance. However, audio does not necessarily benefit the Forward setting. For both models, enabling audio slightly decreases Forward scores and increases the ER rate,

indicating that audio can make models more confident to respond before the proactive timestamp. This suggests a trade-off where richer multimodal signals strengthen Instant and Backward reasoning, but may also encourage more aggressive early answering in Forward-looking questions.

### 4.4. Human Test

To validate the reliability of our LLM-as-a-Judge evaluation, we conduct a human test on a sampled subset. We sample 200 videos, with 50 videos per scenario, and manually annotate model outputs under the same task definitions and rubric as our main evaluation. As shown in Tab. 5, human annotations reproduce the key trends in Tab. 3. Proprietary models consistently outperform open-source models in Overall and across settings. The human test also corroborates our main observation on the Forward task, where performance differences are largely driven by response timing. Doubao-Seed-1.6 achieves the best Forward score with a low ER and the highest PC, while models that perform strongly on Instant and Backward, such as Gemini 3 Pro and Doubao-Seed-1.8, exhibit higher ER, which leads to lower Forward scores. Open-source models remain substantially behind on Forward, largely due to excessive early responses or frequent missing outputs, indicating that timing-sensitive

streaming control is a key bottleneck on PhoStream.

## 5. Conclusion

This paper presents PhoStream, the first mobile-centric streaming benchmark that unifies on-screen and off-screen scenarios for evaluating omnimodal assistants. PhoStream features 5,572 open-ended QA pairs, a scalable Automated Generative Pipeline, a realistic Online Inference Pipeline, and an LLM-as-a-Judge evaluation setup. Across a wide range of models, we observe severe Early Response bias on Forward questions, suggesting that current MLLMs remain unreliable at deciding when to respond in streaming settings. We hope PhoStream serves as a practical testbed for building reliable mobile assistants in real-world streaming scenarios.

## Impact Statement

This paper presents work whose goal is to advance the field of machine learning. We introduce PhoStream, a benchmark that evaluates how well AI models understand streaming videos in mobile-centric scenarios. Our main contribution is identifying a critical weakness in current models. They tend to generate answers before observing necessary visual and audio cues from the video stream. This finding helps researchers build more reliable mobile AI assistants. While continuous video and audio processing may raise privacy concerns in real applications, our work provides an evaluation tool rather than a deployed system. We believe rigorous benchmarking is essential for developing AI assistants that are both capable and trustworthy.

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

# A. Appendix

## A.1. More Statistics of PhoStream

Fig. A.1 summarizes additional Benchmark statistics, including a word cloud and the distribution of question timestamps (Timestamp Question).

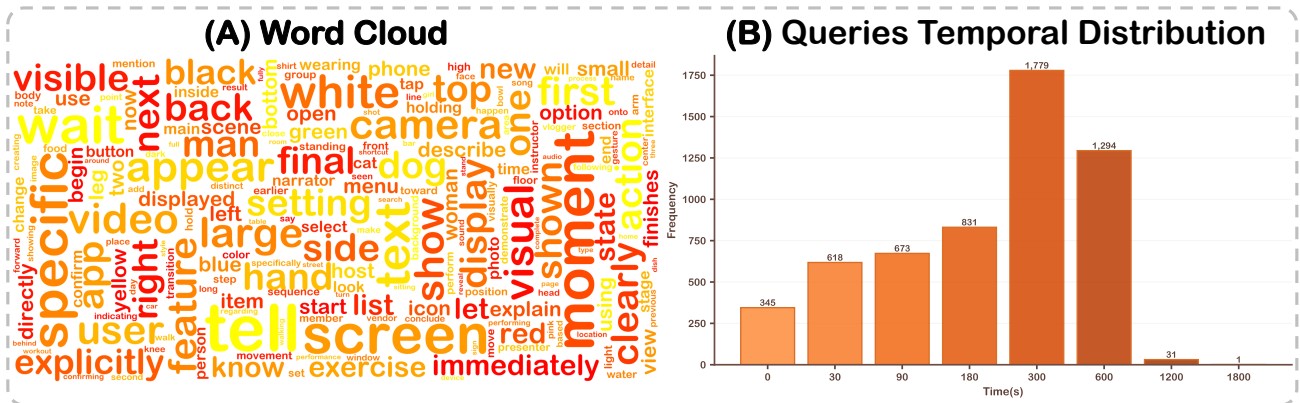

*Figure A.1.* **More QA statistics and distribution.** (A) The word cloud generated from 5,572 open-ended questions illustrates the diverse range of user intents and topics in mobile scenarios. (B) The temporal distribution of queries spans a wide range, with a mean timestamp of 6.52 minutes and a maximum of 30.87 minutes. This significant duration challenges models to maintain long-term context effectively.

## A.2. Details of Human Annotators

In both the benchmark data annotation stage and the human test stage of our experiments, we hire 10 expert annotators with experience in multimodal video understanding, each holding at least a master's degree. Annotators are compensated at a rate of $20 per hour.

## A.3. More Detailed Evaluation Results

To provide a more fine-grained view of model performance, we report detailed results broken down by scenario and capability. Tab. A.1 summarizes evaluation scores across scenario-based dataset subsets (YouTube Vlog, Phone Tutorial, Phone Record, and EgoBlind) under the Instant, Backward, and Forward settings. Tab. A.2, Tab. A.3, and Tab. A.4 report the corresponding breakdown across capability categories.

*Table A.1.* **Detailed evaluation results across different datasets.** We report evaluation scores for Instant, Backward, and Forward tasks on four dataset categories: YouTube Vlog, Phone Tutorial, Phone Record, and EgoBlind. The best result in each category is **bolded**.

| Model | Param | YouTube Vlog | | | Phone Tutorial | | | Phone Record | | | EgoBlind | | |
|---|---|---|---|---|---|---|---|---|---|---|---|---|---|
| | | Inst. | Back. | Forw. | Inst. | Back. | Forw. | Inst. | Back. | Forw. | Inst. | Back. | Forw. |
| *Proprietary Multimodal Models* | | | | | | | | | | | | | |
| Gemini 3 Pro | - | **80.04** | **80.83** | 14.53 | 85.13 | **89.02** | 20.59 | 72.13 | 73.59 | 15.41 | **84.34** | 84.14 | 19.01 |
| Doubao-Seed-1.6 | - | 68.39 | 53.20 | **41.18** | 77.31 | 77.41 | **50.67** | 72.80 | 73.91 | **40.54** | 73.01 | 72.25 | **53.27** |
| Doubao-Seed-1.8 | - | 78.62 | 70.46 | 28.65 | **85.76** | 87.06 | 43.18 | **77.47** | **84.69** | 34.53 | 81.20 | **85.23** | 37.33 |
| *Open-source Multimodal Models* | | | | | | | | | | | | | |
| Qwen2.5-Omni-7B | 7B | 63.35 | 59.65 | 2.28 | 76.64 | 75.53 | 0.65 | 70.13 | 71.25 | 1.82 | 70.84 | 65.95 | 1.78 |
| Qwen3-VL-8B | 8B | 73.14 | 66.07 | **6.28** | 79.95 | 80.94 | **7.19** | **74.53** | 73.12 | **9.93** | 77.59 | **78.74** | **10.40** |
| Qwen3-VL-30B-A3B | 30B | 72.24 | 65.15 | 5.17 | 77.22 | 76.78 | 5.14 | 69.87 | 71.56 | 4.53 | 74.22 | 74.59 | 7.33 |
| Qwen3-Omni-30B-A3B | 30B | **75.91** | **74.88** | 1.19 | **82.06** | **83.14** | 1.28 | 73.73 | **76.25** | 1.49 | 74.22 | 78.20 | 1.49 |
| *Open-source Multimodal Streaming Models* | | | | | | | | | | | | | |
| Dispider | 7B | **47.50** | **45.31** | **4.52** | **39.76** | **42.04** | **1.75** | **34.13** | **36.25** | **2.43** | **45.78** | **38.92** | **2.97** |
| VideoLLM-online-8B | 8B | 26.82 | 27.44 | 0.00 | 21.63 | 21.41 | 0.00 | 21.47 | 18.44 | 0.00 | 24.10 | 24.14 | 0.00 |
| MMDuet2 | 3B | 10.19 | 8.81 | 1.81 | 5.42 | 7.45 | 1.01 | 4.67 | 2.81 | 0.07 | 15.66 | 13.69 | 1.19 |

*Table A.2.* **Detailed evaluation results across different capabilities.** We report evaluation scores for Instant, Backward, and Forward tasks on ten capability categories. The best result in each category is **bolded**.

| Model | Param | Action & Activity | | | Audio-Visual | | | Entity & Knowledge | | | Fine-grained Visual | | |
|---|---|---|---|---|---|---|---|---|---|---|---|---|---|
| | | Inst. | Back. | Forw. | Inst. | Back. | Forw. | Inst. | Back. | Forw. | Inst. | Back. | Forw. |
| *Proprietary Multimodal Models* | | | | | | | | | | | | | |
| Gemini 3 Pro | - | **78.59** | **78.94** | 10.73 | **84.03** | **84.94** | 26.64 | **95.00** | **83.33** | 16.67 | 81.18 | 78.11 | 14.86 |
| Doubao-Seed-1.6 | - | 68.82 | 65.61 | **38.94** | 44.15 | 44.10 | **36.70** | 82.14 | 57.78 | **57.78** | 74.89 | 71.50 | **44.83** |
| Doubao-Seed-1.8 | - | 76.23 | 77.88 | 25.25 | 68.18 | 67.82 | 31.23 | 92.86 | 77.78 | 50.00 | **81.52** | **80.79** | 30.09 |
| *Open-source Multimodal Models* | | | | | | | | | | | | | |
| Qwen2.5-Omni-7B | 7B | 57.38 | 58.79 | 1.53 | 66.67 | 64.55 | 2.01 | 90.71 | 55.56 | 4.44 | 63.49 | 65.20 | 1.71 |
| Qwen3-VL-8B | 8B | **70.80** | **73.18** | **4.70** | 69.06 | 63.65 | **6.68** | 90.00 | 66.67 | **5.56** | 74.74 | 71.97 | **8.75** |
| Qwen3-VL-30B-A3B | 30B | 67.80 | 68.94 | 3.92 | 67.80 | 61.03 | 5.15 | **95.00** | 75.56 | 0.00 | 74.74 | **75.12** | 7.09 |
| Qwen3-Omni-30B-A3B | 30B | 70.29 | 70.30 | 0.58 | **79.75** | **80.00** | 2.66 | 91.43 | **77.78** | 0.00 | **77.59** | 72.91 | 1.16 |
| *Open-source Streaming Models* | | | | | | | | | | | | | |
| Dispider | 7B | **46.07** | **47.73** | **3.65** | **36.23** | **36.86** | **2.37** | **61.43** | **42.22** | **1.11** | **52.73** | **51.02** | **6.18** |
| VideoLLM-online | 8B | 28.18 | 28.33 | 0.00 | 23.02 | 22.50 | 0.00 | 25.71 | 15.56 | 0.00 | 27.57 | 29.61 | 0.00 |
| MMDuet2 | 3B | 12.08 | 8.03 | 1.47 | 4.15 | 5.71 | 1.01 | 5.00 | 0.00 | 0.00 | 11.65 | 13.23 | 0.80 |

*Table A.3.* **Detailed evaluation results across different capabilities (continued).** We report evaluation scores for Instant, Backward, and Forward tasks on the remaining capability categories. The best result in each category is **bolded**.

| Model | Param | Physical & Causal | | | Scene & Context | | | Social & Emotional | | | Sequential & Procedural | | |
|---|---|---|---|---|---|---|---|---|---|---|---|---|---|
| | | Inst. | Back. | Forw. | Inst. | Back. | Forw. | Inst. | Back. | Forw. | Inst. | Back. | Forw. |
| *Proprietary Multimodal Models* | | | | | | | | | | | | | |
| Gemini 3 Pro | - | 64.76 | **84.91** | 7.36 | 87.42 | 86.35 | 17.66 | 74.34 | **79.75** | 9.55 | 82.67 | 76.50 | 14.77 |
| Doubao-Seed-1.6 | - | 55.24 | 69.12 | **41.23** | 78.39 | 76.51 | **47.50** | 64.15 | 60.00 | **31.82** | 72.00 | 67.57 | **52.52** |
| Doubao-Seed-1.8 | - | **72.86** | 81.05 | 24.98 | **89.68** | **89.84** | 34.53 | **75.85** | 76.79 | 20.00 | **90.67** | **76.70** | 44.49 |
| *Open-source Multimodal Models* | | | | | | | | | | | | | |
| Qwen2.5-Omni-7B | 7B | 48.10 | 63.86 | 1.92 | 74.19 | 73.33 | 4.06 | 56.60 | 64.69 | 0.00 | 70.67 | 56.70 | 1.31 |
| Qwen3-VL-8B | 8B | 59.05 | 75.09 | **6.36** | **85.48** | **83.81** | **14.69** | 71.32 | 72.84 | **3.41** | 78.67 | 69.90 | **7.85** |
| Qwen3-VL-30B-A3B | 30B | **61.90** | **78.95** | 4.98 | 80.65 | 81.27 | 4.69 | 68.30 | 69.63 | 2.95 | 80.00 | 66.80 | 4.86 |
| Qwen3-Omni-30B-A3B | 30B | 57.62 | **78.95** | 1.15 | 83.55 | 81.90 | 1.72 | **72.45** | **78.27** | 0.91 | **84.00** | **70.29** | 0.00 |
| *Open-source Streaming Models* | | | | | | | | | | | | | |
| Dispider | 7B | **39.05** | **51.23** | **5.44** | **60.65** | **57.14** | **6.09** | **50.19** | **44.69** | 1.82 | **40.00** | **35.73** | **9.35** |
| VideoLLM-online | 8B | 21.90 | 29.47 | 0.00 | 30.00 | 30.48 | 0.00 | 30.19 | 24.20 | 0.00 | 18.67 | 25.24 | 0.00 |
| MMDuet2 | 3B | 4.29 | 1.75 | 2.53 | 18.39 | 18.41 | 2.66 | 8.30 | 9.63 | **2.73** | 8.00 | 8.54 | 1.87 |

*Table A.4.* **Detailed evaluation results on UI & Digital Interface and Visual Text Understanding.** We report evaluation scores for Instant, Backward, and Forward tasks. The best result in each category is **bolded**.

| Model | Param | UI & Digital Interface | | | Visual Text (OCR) | | |
|---|---|---|---|---|---|---|---|
| | | Inst. | Back. | Forw. | Inst. | Back. | Forw. |
| *Proprietary Multimodal Models* | | | | | | | |
| Gemini 3 Pro | - | 81.85 | **86.70** | 16.90 | 79.89 | 70.00 | 26.52 |
| Doubao-Seed-1.6 | - | 75.49 | 78.25 | **51.89** | 84.04 | 66.11 | **59.28** |
| Doubao-Seed-1.8 | - | **83.38** | 85.67 | 43.56 | **87.08** | **75.00** | 48.12 |
| *Open-source Multimodal Models* | | | | | | | |
| Qwen2.5-Omni-7B | 7B | 76.26 | 73.61 | 0.55 | 79.55 | 67.22 | 7.25 |
| Qwen3-VL-8B | 8B | 77.79 | 78.97 | **7.65** | **82.70** | 68.33 | **10.29** |
| Qwen3-VL-30B-A3B | 30B | 75.13 | 74.85 | 5.38 | 78.99 | 63.89 | 9.86 |
| Qwen3-Omni-30B-A3B | 30B | **80.56** | **81.65** | 0.98 | 79.66 | **76.67** | 1.88 |
| *Open-source Streaming Models* | | | | | | | |
| Dispider | 7B | **38.05** | **41.44** | **1.44** | **33.71** | **35.56** | 1.59 |
| VideoLLM-online | 8B | 21.54 | 19.90 | 0.00 | 19.66 | 26.11 | 0.00 |
| MMDuet2 | 3B | 4.21 | 7.53 | 0.89 | 8.88 | 11.67 | **1.88** |

*Table A.5.* **Effect of context window length on Backward questions.** We evaluate Gemini 3 Pro with different context window lengths. Longer windows improve Backward performance, while the gains diminish as the window becomes sufficiently large.

| Context Window | Backward Score (↑) | Gain |
|:---:|:---:|:---:|
| 30s | 79.8 | – |
| 60s | 82.2 | **+2.4** |
| 120s | **83.1** | +0.9 |

## A.4. Effect of Context Window on Backward Questions

We further study how the context window length affects Backward questions using Gemini 3 Pro. As shown in Tab. A.5, increasing the context window consistently improves Backward performance. The score increases from 79.8 with a 30s window to 82.2 with a 60s window, and further to 83.1 with a 120s window. However, the improvement shows diminishing returns. Extending the window from 30s to 60s brings a gain of 2.4 points, while extending it from 60s to 120s yields only 0.9 points. This suggests that longer temporal context helps models recover relevant past evidence for Backward questions, but a moderate-length window already captures most of the useful information.

## A.5. Further Analysis of the Audio Modality

Tab. 4 shows that audio improves Instant and Backward performance, but slightly reduces Forward performance. This indicates that audio provides useful complementary evidence for understanding observed content, while also making models more willing to respond. For Forward questions, audio cues may appear earlier than the decisive visual evidence and provide only partial semantic information, which can encourage premature hypotheses and early responses. This explains the simultaneous increase in Early Response and decrease in No Response after enabling audio. Thus, audio benefits multimodal comprehension, but may amplify over-confident response triggering in timing-sensitive streaming settings.

## A.6. Failure Cases and Qualitative Results on PhoStream

Fig. A.2 shows that many models answer before the required visual evidence appears in the stream. We report each model's first non-silent, non-placeholder response time under our evaluation protocol and the corresponding response content. Under strict cutoffs, response timing becomes a key factor for correctness in streaming QA.

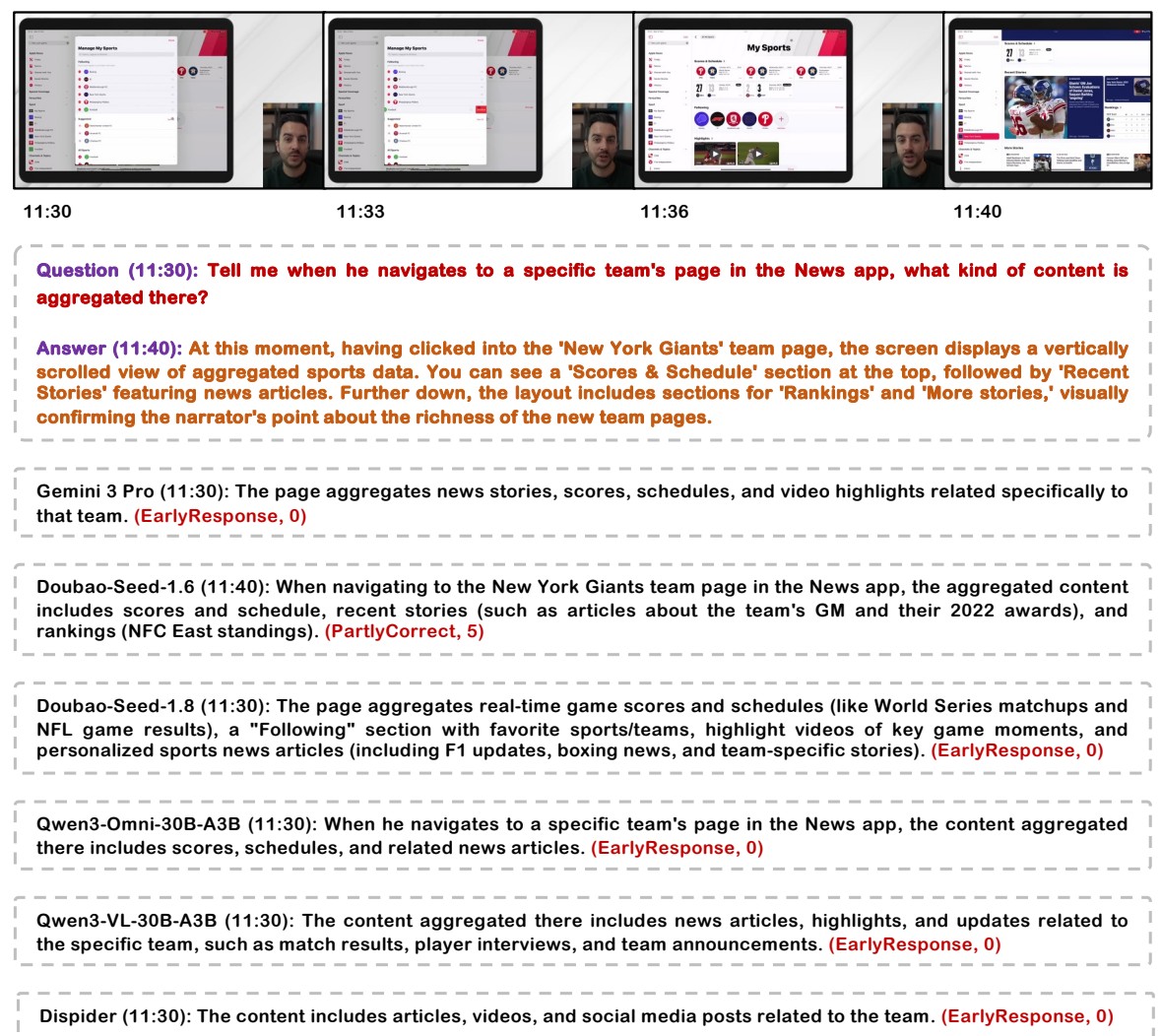

**Question (11:30): Tell me when he navigates to a specific team's page in the News app, what kind of content is aggregated there?**

**Answer (11:40): At this moment, having clicked into the 'New York Giants' team page, the screen displays a vertically scrolled view of aggregated sports data. You can see a 'Scores & Schedule' section at the top, followed by 'Recent Stories' featuring news articles. Further down, the layout includes sections for 'Rankings' and 'More stories,' visually confirming the narrator's point about the richness of the new team pages.**

**Gemini 3 Pro (11:30):** The page aggregates news stories, scores, schedules, and video highlights related specifically to that team. **(EarlyResponse, 0)**

**Doubao-Seed-1.6 (11:40):** When navigating to the New York Giants team page in the News app, the aggregated content includes scores and schedule, recent stories (such as articles about the team's GM and their 2022 awards), and rankings (NFC East standings). **(PartlyCorrect, 5)**

**Doubao-Seed-1.8 (11:30):** The page aggregates real-time game scores and schedules (like World Series matchups and NFL game results), a "Following" section with favorite sports/teams, highlight videos of key game moments, and personalized sports news articles (including F1 updates, boxing news, and team-specific stories). **(EarlyResponse, 0)**

**Qwen3-Omni-30B-A3B (11:30):** When he navigates to a specific team's page in the News app, the content aggregated there includes scores, schedules, and related news articles. **(EarlyResponse, 0)**

**Qwen3-VL-30B-A3B (11:30):** The content aggregated there includes news articles, highlights, and updates related to the specific team, such as match results, player interviews, and team announcements. **(EarlyResponse, 0)**

**Dispider (11:30):** The content includes articles, videos, and social media posts related to the team. **(EarlyResponse, 0)**

*Figure A.2.* The question is asked at 11:30, but the user only opens the New York Giants team page at 11:40. The relevant content is visible at 11:40, including Scores & Schedule, Recent Stories, Rankings, and More stories. Most models answer at 11:30 before the evidence appears and are labeled EarlyResponse, while Doubao-Seed-1.6 answers at 11:40 and is partly correct.

Fig. A.3 shows that some models answer before the winner is declared, while others fail to answer at all. We report each model's first non-silent, non-placeholder response time under our evaluation protocol and the corresponding response content. The question explicitly requires waiting until the challenge ends and the winner is announced, which makes premature answers Early Response and missing answers No Response.

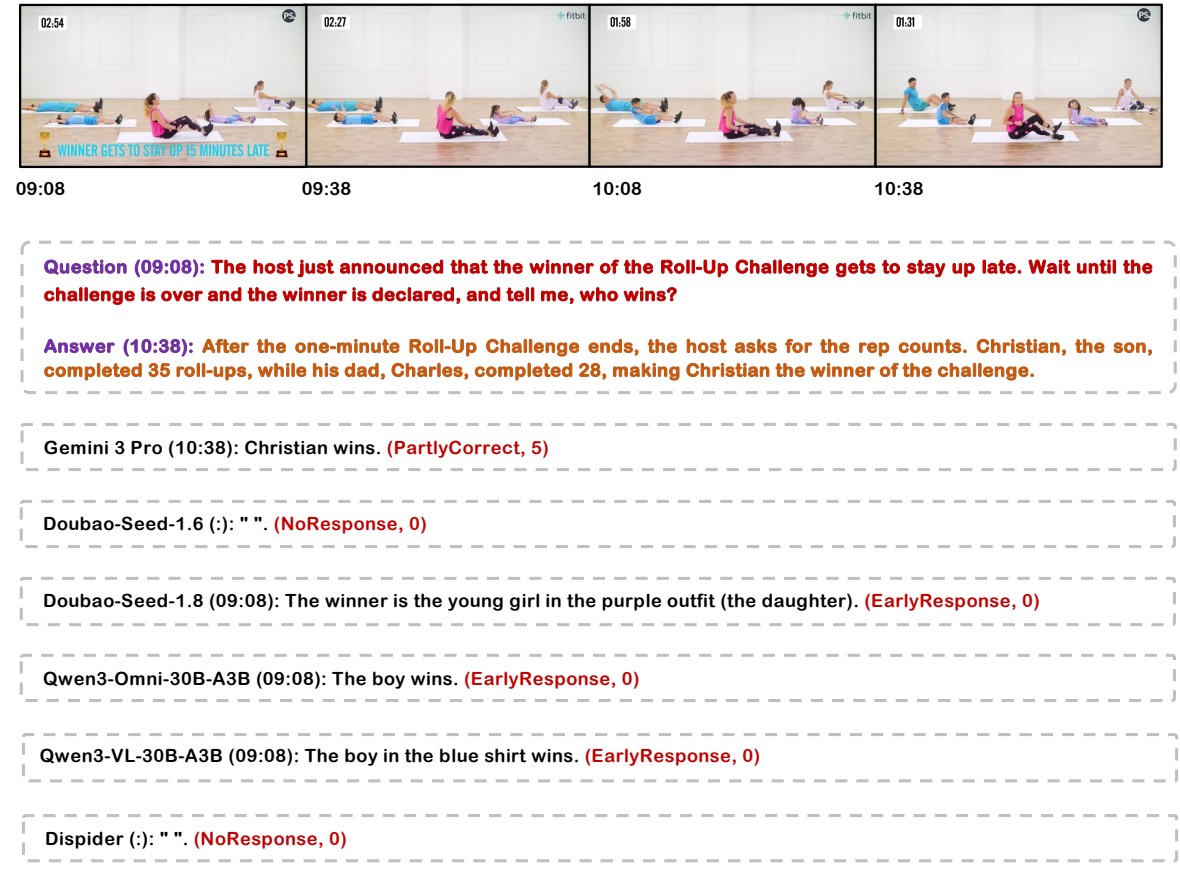

*Figure A.3.* The question is asked at 09:08 and requires waiting until the Roll-Up Challenge ends and the winner is declared. The winner is only known at 10:38, when Christian is announced as the winner after reporting 35 roll-ups versus Charles with 28. Several models answer immediately at 09:08 and are labeled EarlyResponse, while Doubao-Seed-1.6 and Dispider produce no response and are labeled NoResponse.

Fig. A.4 shows that when the required evidence is in the audio stream, models may still fail due to missing or misinterpreting the voiceover content. We report each model's first non-silent, non-placeholder response time under our evaluation protocol and the corresponding response content. From 00:40 to 00:48, the female voiceover complains that her mobile data runs out very quickly and wonders which app is consuming so much traffic; among the models, Gemini 3 Pro captures this complaint correctly, while Qwen3-Omni only partially matches it and others guess or fail.

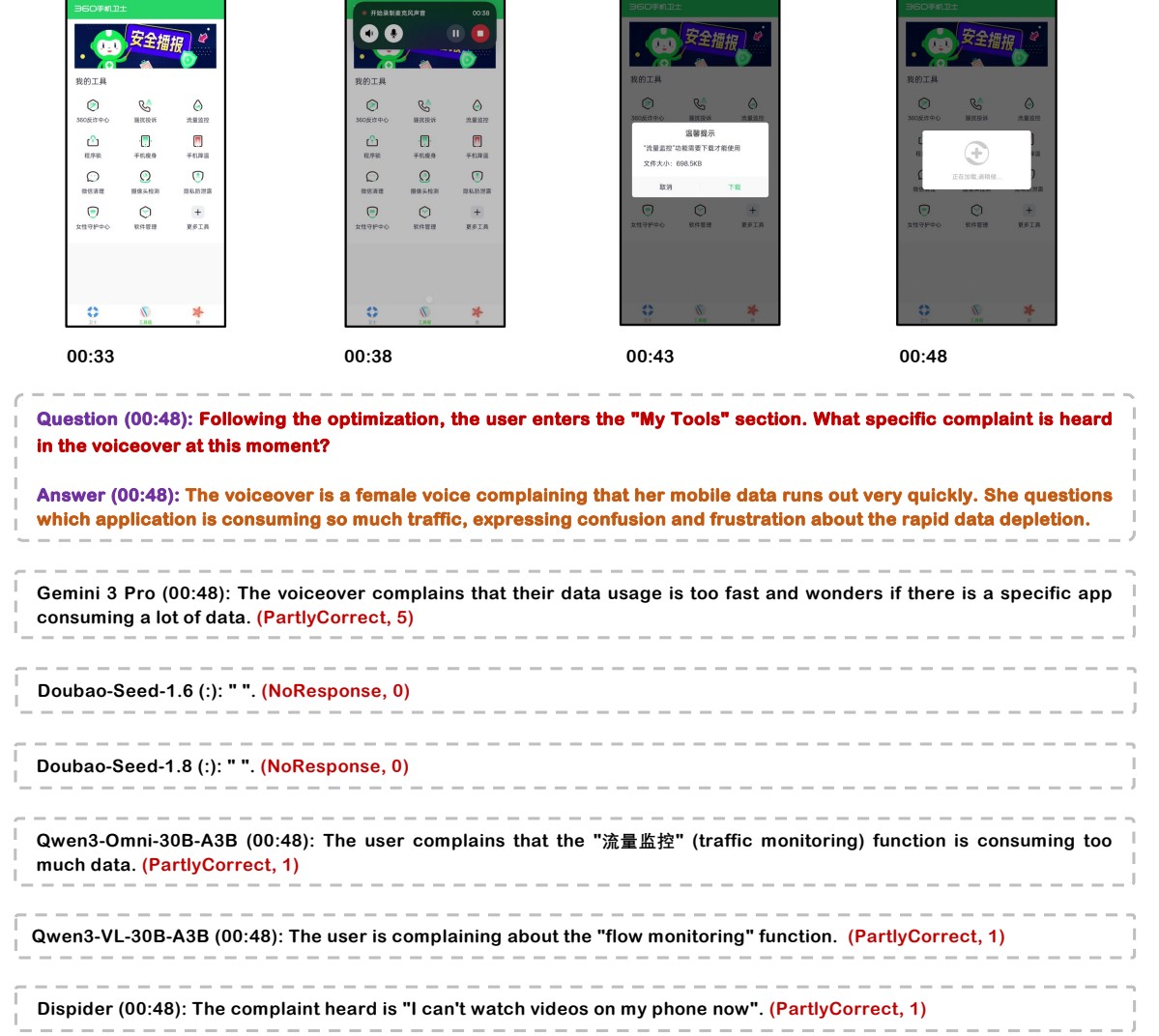

00:33      00:38      00:43      00:48

**Question (00:48): Following the optimization, the user enters the "My Tools" section. What specific complaint is heard in the voiceover at this moment?**

**Answer (00:48): The voiceover is a female voice complaining that her mobile data runs out very quickly. She questions which application is consuming so much traffic, expressing confusion and frustration about the rapid data depletion.**

**Gemini 3 Pro (00:48): The voiceover complains that their data usage is too fast and wonders if there is a specific app consuming a lot of data. (PartlyCorrect, 5)**

**Doubao-Seed-1.6 (:): " ". (NoResponse, 0)**

**Doubao-Seed-1.8 (:): " ". (NoResponse, 0)**

**Qwen3-Omni-30B-A3B (00:48): The user complains that the "流量监控" (traffic monitoring) function is consuming too much data. (PartlyCorrect, 1)**

**Qwen3-VL-30B-A3B (00:48): The user is complaining about the "flow monitoring" function. (PartlyCorrect, 1)**

**Dispider (00:48): The complaint heard is "I can't watch videos on my phone now". (PartlyCorrect, 1)**

*Figure A.4.* The question is asked at 00:48 after the user enters the "My Tools" section. The voiceover complains that mobile data depletes very quickly and asks which application is using so much traffic. Gemini 3 Pro answers correctly at 00:48, whereas Qwen3-Omni responds at 00:48 but misattributes the complaint to a specific function (e.g., "traffic monitoring"), and non-omni models output incorrect guesses or no response.

Fig. A.5 shows that questions requiring aggregation over the whole stream can be challenging, even when models respond at the correct time. We report each model's first non-silent, non-placeholder response time under our evaluation protocol and the corresponding response content. Several models identify parked cars as the dominant obstacle, while others focus on salient background objects such as trees.

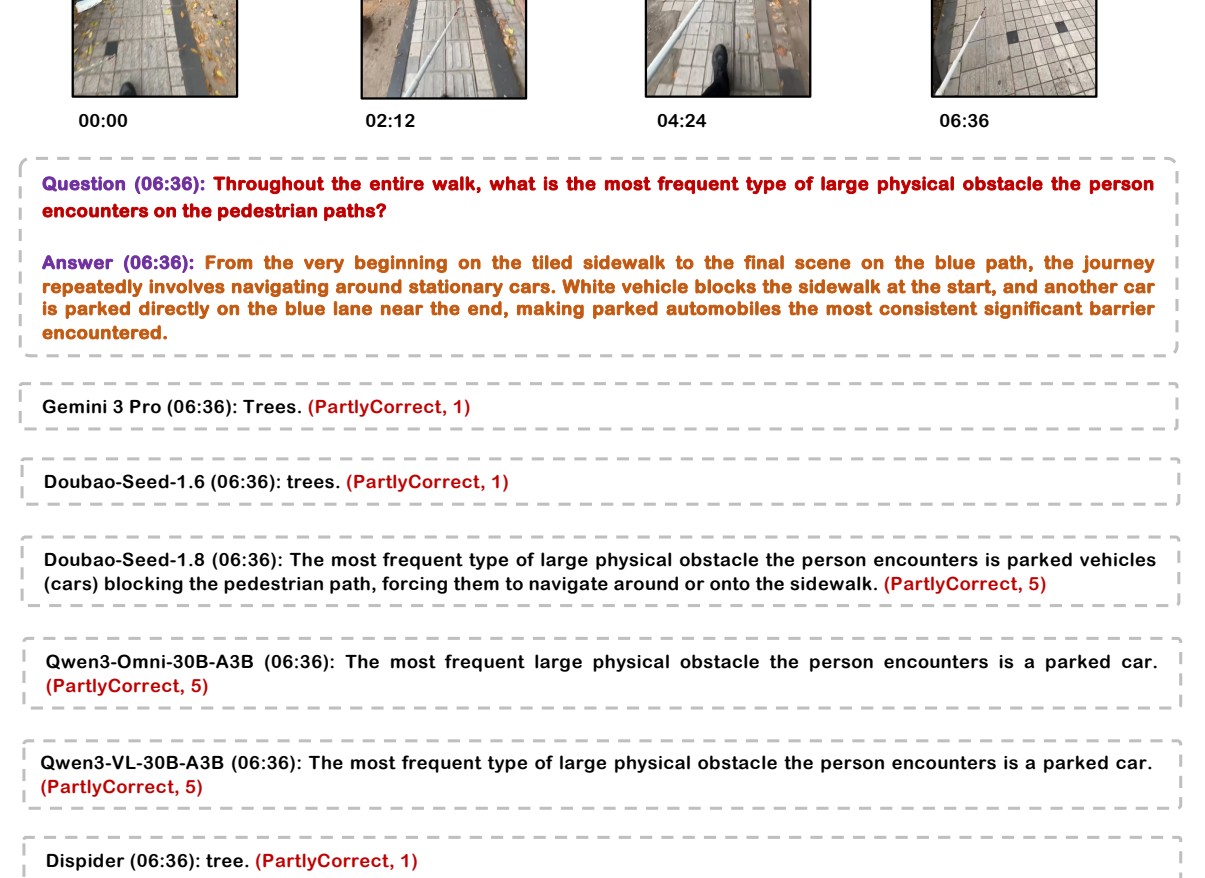

00:00        02:12        04:24        06:36

**Question (06:36): Throughout the entire walk, what is the most frequent type of large physical obstacle the person encounters on the pedestrian paths?**

**Answer (06:36): From the very beginning on the tiled sidewalk to the final scene on the blue path, the journey repeatedly involves navigating around stationary cars. White vehicle blocks the sidewalk at the start, and another car is parked directly on the blue lane near the end, making parked automobiles the most consistent significant barrier encountered.**

**Gemini 3 Pro (06:36): Trees. (PartlyCorrect, 1)**

**Doubao-Seed-1.6 (06:36): trees. (PartlyCorrect, 1)**

**Doubao-Seed-1.8 (06:36): The most frequent type of large physical obstacle the person encounters is parked vehicles (cars) blocking the pedestrian path, forcing them to navigate around or onto the sidewalk. (PartlyCorrect, 5)**

**Qwen3-Omni-30B-A3B (06:36): The most frequent large physical obstacle the person encounters is a parked car. (PartlyCorrect, 5)**

**Qwen3-VL-30B-A3B (06:36): The most frequent type of large physical obstacle the person encounters is a parked car. (PartlyCorrect, 5)**

**Dispider (06:36): tree. (PartlyCorrect, 1)**

*Figure A.5.* The question is asked at 06:36 and requires summarizing the entire walk. Across the video, parked cars repeatedly obstruct the pedestrian path, making vehicles the most frequent large physical obstacle. Some models answer cars, while others answer trees, which leads to lower scores.

## A.7. Placeholder Responses for Filtering in Evaluation

*Listing A.1.* Placeholder responses used for filtering in evaluation.

```
PLACEHOLDERS = {
    "", "silent", "<NO_INFORMATION>", "<SILENT>",
    "got it, i'll let you know.",
    "收到，我会留意的。",
    "没问题，到时候提醒你。",
    "好的，到时候我会提醒你。",
    "没问题，到时候我会告诉你。",
    "No problem, I'll remind you then.",
    "Okay, I will alert you when it happens.",
    "I will make sure to remind you at that time.",
    "好的，那到时候我会提醒你。",
    "好的，到时候我会发提醒给你。",
    "Sure, I will let you know at that time.",
    "Noted, expect a reminder from me then.",
    "Ok, I will remind you then.",
    "Certainly, I will provide the reminder then.",
    "No problem, I'll remind you then.",
    "ok", "okay", "sure", "yes", "收到", "好的", "明白了",
    "got it, i will notify you at that moment."
}
```

## A.8. Evaluation Rubric for LLM-as-a-Judge and Human Test

*Listing A.2.* Evaluation prompt for LLM-as-a-Judge and Human Test.

```
prompt = '''You are an expert evaluator judging whether a model's answer provides a reasonable and factually
plausible explanation that directly addresses the question, based on the reference answer.

**Evaluation Guideline:**
- Focus on whether the model gives a coherent reason that logically explains what the question asks.
- The answer does not need to reproduce all details from the reference - it only needs to offer a factually grounded
 and relevant cause.
- An answer that captures the essential reason should be considered strong, even if it omits descriptive details.
- Accept simplified, rephrased, or high-level reasoning as long as it is consistent with the reference, plausibly
explains the phenomenon in the question, and does not contradict known facts.
- Do not deduct points for omitting secondary or illustrative details when the core causal logic is present, or for
using concise or abstract phrasing.
- Only penalize if the explanation is factually wrong, fails to provide a meaningful cause, or is so vague that it
does not actually answer the question.

**Scoring (integer 0-5):**
- 5: Fully accurate and complete explanation.
- 4: Correct and logically sufficient explanation; may omit non-essential details but captures the essential reason.
- 3: Partially relevant but weakens or misses part of the core causal link.
- 2: Tangential or speculative without solid grounding.
- 1: Factually incorrect.
- 0: No attempt to answer or completely off-topic.

**Output Format:**
Return a valid JSON object with exactly two keys:
- "explanation": one sentence focusing on whether the answer gives a reasonable and relevant reason for the question
- "score": an integer from 0 to 5

Output only the JSON. No other text, markdown, or commentary.

**Inputs:**
- Question: {question}
- Predicted Answer: {model_output}
- Correct Answer: {reference_answer}
'''
```

