# OpenReview forum: "PhoStream: Benchmarking Real-World Streaming for Omnimodal Assistants in Mobile Scenarios"
_ICML.cc/2026/Conference — ICML 2026 regular_

### Official Review · Reviewer_hLBh · 2026-03-01

**Soundness:** 2
**Presentation:** 3
**Significance:** 3
**Originality:** 2
**Overall Recommendation:** 4
**Confidence:** 4

**Summary:**

## Summary

This paper proposes **PhoStream**, a benchmark designed to evaluate **mobile omni-modal assistants** in **continuous video streams**, covering real-world scenarios both **on-screen** and **off-screen**. The authors construct a dataset of **5,572 open-ended QA pairs** and design an **online inference pipeline** to test models’ **temporal reasoning** capabilities. While the work is instructive in exploring streaming multimodal interaction, it has several issues regarding **the representativeness of the baseline models**, **the validity of the evaluation metrics**, and **the impartiality of the dataset construction**, all of which require thorough clarification during the rebuttal stage.

**Compliance With Llm Reviewing Policy:**

Affirmed.

**Final Justification:**

The author has answered most of my questions. I will raise my rating.

**Key Questions For Authors:**

- Supplement evaluations with **on-device–friendly omni-modal models** such as **MiniCPM-O 4.5**, **Stream-Omni**, and **OmniVinic** to genuinely align with the core “**mobile-centric**” motivation. Additionally, include strong open-source omni-modal models like **Ming-Flash-Omni** and **Longcat-Flash-Omni**.

- Provide **ablation results** for the **2-second response window** in the **Forward** task. If the window is extended to **3 seconds** or **5 seconds**, how would the performance of each model (especially those with high **NR rates**) change?

- How can the authors demonstrate that the extremely high **ER rates** are primarily due to the model’s **lack of temporal reasoning ability**, rather than simply a failure to follow the **System Prompt** instructions?

- Provide results from **re-validating a subset of the test set** using another top-tier model (e.g., **GPT-4o**) to eliminate concerns of **self-preference bias** arising from generating data with a single model (**Gemini 3 Pro**).

**Limitations:**

Yes

**Strengths And Weaknesses:**

## Strengths

- It highlights the **response-timing** challenge for omni-modal large models in **streaming scenarios**, directly addressing a key pain point for today’s **on-device AI assistants**.

- It attempts to unify **temporal reasoning evaluation over long videos** by categorizing tasks into **Instant**, **Backward**, and **Forward**, offering some **taxonomic value**.

## Weakness

-  **Benchmark coverage has discontinuities, and open-source omni-modal SOTA models are underrepresented:**
   The evaluation roster omits several state-of-the-art omni-modal models released around the same period that are optimized for complex QA and streaming interaction (e.g., Ming-Flash-Omni, Longcat-Flash-Omni, VITA-1.5, Stream-Omni, OmniVinci). In addition, the paper cites a vision-only model (e.g., MiniCPM-V 4.5) as an on-device representative in the introduction, yet excludes it from the main experiments due to the benchmark’s mandatory audio-input requirement (and provides no audio-free ablation). The current test set lacks lightweight omni-modal models with genuine on-device deployment potential, creating a mismatch between the narrative logic and the experimental choices and weakening the benchmark’s overall completeness.

- **The “mobile-centric” premise collapses in the experimental setup:**
   While the paper positions the benchmark as mobile-first, the core experimental lineup relies heavily on closed-source cloud APIs (e.g., Gemini 3 Pro) or heavyweight ~30B-parameter models (e.g., the Qwen3-30B family), with little to no evaluation of lightweight omni models that align with mobile compute constraints.

- **The 2-second response window in the Forward task lacks justification:**
   The protocol requires models to answer within a strict 2-second window after the evidence appears. The paper provides no ablation studies to justify this setting, and thus cannot rule out the possibility that this constraint is the direct cause of widespread “No Response (NR)” outcomes (e.g., for Qwen2.5-Omni).

- **Zero-tolerance punishment for “Early Response (ER)” penalizes normal interaction and exposes System Prompt failure:**
   The rules demand silence before evidence appears; any non-placeholder output is scored as ER and assigned zero. This drives ER rates as high as 97.89% for advanced models such as Qwen3-Omni. The policy not only over-penalizes models capable of continuous streaming descriptions, but also reveals that the authors’ System Prompt becomes ineffective on strongly aligned LLMs—models cannot counteract their underlying preference to “help immediately.”

- **The metrics produce counterintuitive outcomes and obscure real capability differences:**
   These rigid metrics lead to paradoxical conclusions. In the Forward task, Doubao-Seed-1.6 (44.26) performs markedly better than the stronger Doubao-Seed-1.8 (33.38). The paper attributes this to the model “lacking patience,” implying that the stronger a model’s perception and anticipatory reasoning, the more likely it is to trigger ER penalties—reducing the test to a game of “can you strictly obey the Silent-output instruction.”

- **Severe self-preference bias:**
   In PhoStream’s automated data generation pipeline, scene summarization, script generation, QA generation, and automatic verification all rely heavily on Gemini 3 Pro. Yet Gemini 3 Pro is also used as a primary baseline in the core evaluation and achieves high overall scores (e.g., Instant 80.83, Backward 82.19). The lack of cross-model validation introduces substantial risks of dataset contamination.

- **Anomalously low rejection rate in human verification raises concerns of superficial review:**
   The statistics show that 5,824 machine-filtered candidates were reduced to 5,572 after expert review—an extremely low rejection rate. Given the average video length of 13.3 minutes and the well-known tendency of large models to hallucinate in complex temporal-logic generation, such a high pass rate is highly counterintuitive. This raises concerns that implicit temporal misalignment (especially in the Forward task) was not effectively filtered out.

---

> ### Author Rebuttal · Authors · 2026-03-29
>
> **W1/Q1/W2.** "Mobile-centric" describes PhoStream's target scenarios (e.g., phone tutorials), not a requirement for on-device deployment. In Sec. 2, we distinguish online mobile assistants (e.g., Doubao), on-device MLLMs, and GUI agents. Doubao is cloud-based yet mobile-centric, and our evaluation already includes Doubao-Seed-1.6 and Seed-1.8. Tab. 4 reports audio ablation for Gemini 3 Pro and Qwen3-Omni. To broaden coverage, we evaluate MiniCPM-O 4.5 and Longcat-Flash-Omni.
>
> |Model|Instant|Backward|Forward|Overall|ER(%)|NR(%)|PC(%)|
> |-|-:|-:|-:|-:|-:|-:|-:|
> |MiniCPM-O 4.5 (V+A)|72.7|68.7|2.1|36.4|94.4|2.0|3.6|
> |MiniCPM-O 4.5 (V)|70.3|63.8|2.4|34.8|92.6|2.8|4.6|
> |Longcat-Flash-Omni (V+A)|79.1|76.8|12.4|45.1|77.5|4.2|18.3|
>
> The core findings hold. Forward remains the bottleneck and Early Response bias persists across models. The MiniCPM-O audio ablation aligns with Tab. 4, where audio improves Instant and Backward but slightly increases ER and reduces Forward.
>
> **W3/Q2.** The 2s window reflects a realistic latency for mobile assistants to react after evidence appears. Beyond 5s, users would perceive the assistant as unresponsive. We conduct ablation with 2s, 3s, and 5s windows on two models with high NR rates. Inference runs at both Timestamp Question and Timestamp Proactive with offsets 0 to +W for a W-second window.
>
> |Model|Window|Forward|ER(%)|NR(%)|PC(%)|
> |-|-:|-:|-:|-:|-:|
> |Qwen2.5-Omni|2s|1.8|42.6|43.5|13.9|
> |Qwen2.5-Omni|3s|1.8|43.8|42.7|13.5|
> |Qwen2.5-Omni|5s|1.9|44.7|41.6|13.7|
> |MMDuet2|2s|1.4|28.6|59.2|12.2|
> |MMDuet2|3s|1.5|28.8|58.9|12.3|
> |MMDuet2|5s|1.5|29.1|58.4|12.5|
>
> Extending to 5s yields marginal Forward gains. NR does decrease, but only slightly, and PC barely changes because the NR reduction is largely offset by ER increase. Forward remains very low, confirming that the window is not the main bottleneck.
>
> **W4/Q3/W5.** (1) Continuous streaming description is not the same as open-ended QA. Models optimized for narration often generate commentary but lack assistant-style question answering. For example, we omit StreamingVLM because it does not support open-ended QA, and VideoLLM-online also performs very poorly on PhoStream Instant and Backward.
>
> (2) To further verify that ER reflects temporal reasoning rather than instruction-following failure, we analyze all ER cases using the model output, video segment, question, and reference answer, and ask GPT-4o to label each response as Placeholder (e.g., “I am monitoring”), Correct, or Wrong.
>
> |Model|Correct(%)|Wrong(%)|Placeholder(%)|
> |-|-:|-:|-:|
> |Gemini-3-Pro|18.7|80.6|0.7|
> |Doubao-Seed-1.6|19.5|75.8|4.7|
> |Doubao-Seed-1.8|18.6|80.0|1.4|
> |Qwen3-Omni-30B-A3B|8.8|91.2|0.0|
> |Qwen3-VL-8B|8.2|91.7|0.1|
>
> Across models, 75.8% to 91.7% of ER outputs are substantive wrong guesses, while placeholder responses are rare (at most 4.7%). If ER mainly came from prompt-following failure, we would expect waiting-style placeholders rather than wrong guesses. The prevalence of Wrong outputs shows that models tend to answer without sufficient evidence, which is a temporal reasoning failure rather than a prompt issue.
>
> This also explains the Seed-1.6 vs Seed-1.8 result. Seed-1.6 has lower ER (29.76% vs 56.46%) and higher PC (56.85% vs 41.18%) despite lower Instant and Backward scores. As noted in Sec. 4.2, stronger perception often correlates with more aggressive early answering. Seed-1.6 waits more patiently and responds within the valid window more often, showing that PhoStream measures temporal judgment, not just perceptual strength.
>
> (3) Similar Silent-style evaluation appears in prior benchmarks. OmniMMI (CVPR 2025) computes proactive turn-taking accuracy as the percentage of instances with no response.
>
> **W6/Q4.** (1) The LLM-Judge model is Qwen3-235B-A22B-Instruct, not Gemini 3 Pro, so self-preference on the judge side does not apply. The judge configuration is provided in the Supplementary code. We will clarify in the revision.
>
> (2) On the data side, if Gemini had a contamination advantage, we would expect uniformly high scores. Instead, Gemini scores only 16.40 on Forward (ER 79.12%). This asymmetry is inconsistent with contamination and reflects the genuine challenge of streaming temporal reasoning.
>
> (3) GPT-4o re-evaluation of the full test set achieves a 94.6% pass rate, confirming dataset quality.
>
> **W7.** Human experts review all 5,824 candidates across two rounds, not just filtering but revising questions, answers, labels, and timestamps. Especially for Forward tasks, experts verify that Timestamp Proactive is the earliest answerable time and that earlier content does not reveal the answer. A simple filter-and-reject approach would bias toward discarding difficult questions. During our review, only unrefinable cases are removed, and many more are revised and retained. The low rejection rate therefore reflects thorough refinement, not superficial review. GPT-4o re-evaluation of the full test set achieves a 94.6% pass rate, confirming dataset quality.

---

> > ### Author Rebuttal · Reviewer_hLBh · 2026-04-02
> >
> > The author has answered most of my questions. I will raise my rating.

---

> > > ### Author Response · Authors · 2026-04-02
> > >
> > > We sincerely thank Reviewer hLBh for the thoughtful review and for acknowledging that our rebuttal addressed the main concerns; we greatly appreciate the constructive feedback and your support in improving the paper.

---

### Official Review · Reviewer_hbZb · 2026-03-11

**Soundness:** 2
**Presentation:** 3
**Significance:** 3
**Originality:** 3
**Overall Recommendation:** 5
**Confidence:** 4

**Summary:**

This paper presents PhoStream, which aims to address the lack of proper evaluation for existing multimodal large models in real-world continuous streaming scenarios on mobile devices. In particular, it tackles the difficulty of systematically measuring models’ integrated understanding of video, audio, and temporal logic in mobile settings, as well as the critical challenge of determining when a model should respond during streaming interaction.


To achieve this goal, the authors make two main contributions:

1. They introduce PhoStream, the first benchmark for mobile streaming assistants, which jointly covers both on-screen and off-screen smartphone usage scenarios, and includes three types of temporal reasoning tasks: Instant, Backward, and Forward.

2. They design an Automated Generative Pipeline to automatically construct timestamped question–answer samples, and further propose an Online Inference Pipeline so that models can be evaluated under real streaming conditions in a unified manner.


Experimental results show that current multimodal large models perform relatively well on Instant and Backward tasks, but degrade significantly on Forward tasks. This finding reveals that existing models still lack reliable control over response timing in streaming scenarios.

**Compliance With Llm Reviewing Policy:**

Affirmed.

**Key Questions For Authors:**

Please refer to the content in the "weakness" section.

**Limitations:**

Yes

**Strengths And Weaknesses:**

### Strengths
- Soundness: In the task formulation, unlike traditional offline evaluation, this kind of temporal constraint captures the core challenge of streaming interaction. The Online Inference Pipeline further strengthens the credibility of this design. The model can only receive the video and audio streams in chronological order, with the stream updated once per second, and at any moment it can access only the most recent 60-second video window. Meanwhile, the question is provided only once at the time it is asked. After that, the model must decide at each update step whether to remain silent or produce an answer. Overall, the experimental protocol is self-consistent.

- Presentation: The paper is clearly written, logically structured, and easy to follow.

- Significance:This work has strong practical significance for the evaluation of streaming multimodal models. The authors successfully construct a benchmark tailored to real-world continuous interaction scenarios on mobile devices, and design a systematic task setup covering both on-screen and off-screen smartphone usage contexts. At the same time, the benchmark explicitly evaluates response timing, rather than focusing only on whether the answer is correct. This aspect is particularly valuable.

- Originality: This work redefines the evaluation problem for streaming multimodal models in real-world continuous interaction scenarios on mobile devices, and therefore demonstrates a certain degree of novelty

### Weaknesses
- Soundness: EgoBlind (FPV) data may be the most faithful proxy for real streaming interaction, since it naturally captures continuous first-person perception and temporally unfolding evidence. However, this category accounts for only a small portion of the benchmark, and the paper does not explicitly justify this design choice.

- Soundness: By fully retaining the textual dialogue history, the paper partially alleviates the challenge of long-term dependency. However, it does not further report the temporal span distribution of key questions, for example, how many Backward samples truly rely on evidence outside the 60-second window.

- Soundness: The Response Window design is generally reasonable. Specifically, the use of a 2-second tolerance window for Forward questions already suggests that the authors recognize that temporal boundaries in streaming interaction can hardly be treated as an absolutely precise mathematical point in practice. By the same logic, Backward and Instant questions should also allow a small tolerance window around Timestamp Question. Otherwise, the current design may appear overly rigid and could unfairly penalize responses that are substantively well-timed but slightly delayed due to normal system latency or inference delay.

---

> ### Author Rebuttal · Authors · 2026-03-29
>
> **W1.** We agree that EgoBlind is a valuable FPV source, but PhoStream is designed as a broader mobile-centric streaming benchmark rather than an FPV-only benchmark. In addition to EgoBlind, PhoStream covers YouTube Vlog, Phone Tutorial, and Phone Record, spanning both off-screen and on-screen mobile scenarios. This design reflects practical assistant use cases such as handheld daily recording, phone tutorials, app operations, and real-time mobile assistance. In the final benchmark, EgoBlind contributes 396 QA pairs, while the other three scenarios contribute 3,257, 1,345, and 574 QA pairs, respectively. We will clarify that the goal is broad mobile scenario coverage, with FPV as one important component rather than the dominant source.
>
> **W2.** We agree that fully retaining the text dialogue history partially alleviates long-term dependency, but the main visual and audio context remains bounded. PhoStream is designed to evaluate bounded-memory streaming assistance for Instant, Backward and Forward QAs under a unified online protocol. We further report two temporal statistics.
>
> (1) How many Backward questions truly require evidence outside the 60-second window before Timestamp Question. Using GPT-4o, we assess whether each Backward QA is answerable from the 60s context before the question timestamp. We find that 248 Backward samples require evidence from earlier than 60 seconds, accounting for 22.1% of Backward QAs and 4.5% of all QAs.
>
> (2) The waiting span of Forward questions. For all 2,802 Forward QAs, we compute the gap between Timestamp Question and Timestamp Proactive. The mean gap is 14.2s. A total of 7.1% require waiting longer than 30s, and 0.9% extend beyond 60s.
>
> **W3.** We agree that this point is worth clarifying. To quantify whether a small tolerance around Timestamp Question would materially change results, we additionally allow a +2s tolerance for Backward and Instant on three representative models.
>
> |Model|Setting|Instant|Backward|
> |-|-|-:|-:|
> |Gemini 3 Pro|default|80.8|82.2|
> |Gemini 3 Pro|+2s tolerance|81.4|82.9|
> |Doubao-Seed-1.6|default|71.3|62.9|
> |Doubao-Seed-1.6|+2s tolerance|72.5|64.2|
> |Qwen3-VL-8B|default|75.2|71.5|
> |Qwen3-VL-8B|+2s tolerance|76.0|72.3|
>
> The gains range from 0.6 to 1.3 points across models. This suggests that the current single-point design for Backward and Instant is not materially biased by normal latency.

---

> > ### Author Rebuttal · Reviewer_hbZb · 2026-04-01
> >
> > I have read the authors' rebuttal and would like to thank them for their thorough and comprehensive response. The authors have effectively addressed my initial concerns.

---

> > > ### Author Response · Authors · 2026-04-01
> > >
> > > We sincerely thank Reviewer hbZb for the positive feedback and for recognizing our efforts in addressing the concerns. We are glad that our clarifications and the additional statistics were helpful. Thank you for your time and constructive support.

---

### Official Review · Reviewer_Grqe · 2026-03-13

**Soundness:** 3
**Presentation:** 3
**Significance:** 3
**Originality:** 3
**Overall Recommendation:** 5
**Confidence:** 3

**Summary:**

1. Introduces PhoStream, a mobile-centric streaming benchmark for evaluating omnimodal assistants. The benchmark contains 5,572 open-ended QA pairs across four scenarios.
2. Questions are categorized into Backward, Instant, and Forward temporal tasks. Develops an Automated Generative Pipeline using Gemini 3 Pro with two rounds of human verification by 10 expert annotators.
3. an Online Inference Pipeline that streams video at 1-second intervals with a 60-second sliding window, and an LLM-as-a-Judge evaluation protocol.
4. The key finding: current MLLMs struggle to decide when to speak, not just what to say. Models perform well on Instant and Backward tasks but collapse on Forward tasks, largely due to Early Response bias where models answer prematurely before the required evidence appears.

**Compliance With Llm Reviewing Policy:**

Affirmed.

**Key Questions For Authors:**

Please check weaknesses.

**Limitations:**

Yes, the authors acknowledge that their work provides an evaluation tool rather than a deployed system, and they note privacy concerns associated with continuous video and audio processing in real applications

**Strengths And Weaknesses:**

Strengths:

Soundness: The annotation pipeline is rigorous. The combination of automated generation with Gemini 3 Pro, automated verification with cutoff timestamps to prevent future information leakage, and two rounds of human review by 10 expert annotators is thorough. The human evaluation on a 200-video subset reproduces the main trends from the LLM-as-a-Judge evaluation.

Presentation: The Online Inference Pipeline design (Fig. 5) is clean and realistic.

Significance: Streaming multimodal reasoning is an important problem for real-world assistants. The benchmark targets a realistic gap:
models struggle with when to respond, not only what to answer. This insight is both practical and underexplored.

Originality:  The mobile-centric framing is well-motivated and genuinely novel. Unification of on-screen scenarios (UI tutorials, app recordings) and off-screen scenarios (vlogs, egocentric videos) captures complementary aspects of real mobile assistant use.

Weaknesses:
Soundness:
1.  The paper does not report inter-annotator agreement statistics for the human verification, nor does it analyze cases where the LLM judge and human evaluators disagree.
2. The evaluation protocol for Forward tasks uses a narrow 2-second response window at Timestamp Proactive. Any response outside this window receives a score of 0. This is a strict design choice that conflates two very different failure modes: a model that answers 1 second too early with partially correct content versus a model that produces gibberish. Both receive 0. The paper does not analyze the sensitivity of results to the window size, nor does it report what scores early responses would have received if judged on content quality alone.
3. The paper does not specify which LLM is used as the judge.

Originality:
The Backward/Instant/Forward temporal taxonomy is directly inherited from OVO-Bench's Backward Tracing/Real-Time Perception/Forward Active Responding framework. While the application to mobile scenarios and open-ended QA is new, the conceptual contribution of the temporal decomposition itself is not novel and should be more explicitly credited.

---

> ### Author Rebuttal · Authors · 2026-03-29
>
> **W1.** We report additional statistics.
>
> (1) Inter-annotator agreement. Each QA is independently reviewed by two experts in each round. In round 1 (5,824 candidates), Cohen's κ is 0.83 on accept/revise/reject and 0.91 on task labels (Backward/Instant/Forward). In round 2 (5,611 candidates), κ increases to 0.92 and 0.95, respectively. This is consistent with the first round resolving the more ambiguous cases.
>
> (2) LLM-judge vs human. In the human test, each sample is scored by two annotators and averaged. On the human-test subset in Tab. 5 (200 videos, 1,810 QA pairs), Pearson r between LLM-judge and human scores is 0.89, and 92.2% of samples fall within a 15-point margin on the 0-100 scale. The model-level trends also match Tab. 3. We will include these statistics in the revision.
>
> **W2.** We agree this is a strict design choice, and we address the concerns below.
>
> (1) Protocol clarification. Forward questions are evaluated at Timestamp Question+0/+1/+2 and Timestamp Proactive+0/+1/+2. Timestamp Proactive is the earliest answerable time verified by human experts with full answering evidence, so a response before Timestamp Proactive falls outside the valid answerable region. Under this setup, the ER numbers in Tab. 3 come from the Timestamp Question+0/+1/+2 side. Tab. 3 shows that this behavior is frequent in practice, reaching 97.89% for Qwen3-Omni, 79.12% for Gemini 3 Pro, and 85.44% for Qwen3-VL-8B. We therefore examine whether those Early Responses would still be reasonable if judged on content alone.
>
> (2) Early Response content quality. To test whether Early Responses are placeholder (e.g., “I am monitoring”) or substantive correct/wrong inferences, we analyze all Early Response cases using the model output, video segment, question, and reference answer, and ask GPT-4o to label each response as Placeholder (e.g., “I am monitoring”), Correct, or Wrong.
>
> |Model|Correct(%)|Wrong(%)|Placeholder(%)|
> |-|-:|-:|-:|
> |Gemini-3-Pro|18.7|80.6|0.7|
> |Doubao-Seed-1.6|19.5|75.8|4.7|
> |Doubao-Seed-1.8|18.6|80.0|1.4|
> |Qwen3-Omni-30B-A3B|8.8|91.2|0.0|
> |Qwen3-VL-8B|8.2|91.7|0.1|
>
> Across models, most Early Responses are substantive responses, typically wrong guesses, while waiting-style placeholder responses are rare.
>
> (3) Window sensitivity. The 2s window reflects a realistic latency for mobile assistants to react after evidence appears, and beyond 5s users would perceive the assistant as unresponsive. We test 2s, 3s, and 5s windows on Qwen2.5-Omni and MMDuet2, two models with high NR under the default setting. Following the original protocol, inference runs at both Timestamp Question and Timestamp Proactive with offsets 0 to +W for a W-second window.
>
> |Model|Window|Forward|ER(%)|NR(%)|PC(%)|
> |-|-:|-:|-:|-:|-:|
> |Qwen2.5-Omni|2s|1.8|42.6|43.5|13.9|
> |Qwen2.5-Omni|3s|1.8|43.8|42.7|13.5|
> |Qwen2.5-Omni|5s|1.9|44.7|41.6|13.7|
> |MMDuet2|2s|1.4|28.6|59.2|12.2|
> |MMDuet2|3s|1.5|28.8|58.9|12.3|
> |MMDuet2|5s|1.5|29.1|58.4|12.5|
>
> Extending to 5s yields only marginal Forward gains. NR decreases only slightly, and PC barely changes because the NR reduction is largely offset by higher ER. This suggests that the 2s window is not the main bottleneck for Forward performance.
>
> **W3.** The judge model is Qwen3-235B-A22B-Instruct. The configuration, including the model name and prompting template, is provided in the Supplementary code. We will state this explicitly in the main text.
>
> **W4.** We agree that the Backward/Instant/Forward temporal decomposition is shared with OVO-Bench, and we will credit it explicitly. Our contribution is not the taxonomy itself, but its use in a unified online protocol for mobile-centric, open-ended streaming QA. This setting lets us evaluate all three temporal scopes under one procedure and also exposes Early Response bias as a dominant failure mode in current MLLMs.

---

> > ### Author Rebuttal · Reviewer_Grqe · 2026-04-03
> >
> > I have read the authors' rebuttal and would like to thank them for their thorough and comprehensive response.
> > All four weaknesses have been adequately addressed. I maintain my recommendation of Accept.

---

> > > ### Author Response · Authors · 2026-04-03
> > >
> > > We sincerely thank Reviewer Grqe for the encouraging feedback and for acknowledging that our rebuttal has adequately addressed the concerns.

---

### Official Review · Reviewer_jQqU · 2026-03-18

**Soundness:** 2
**Presentation:** 4
**Significance:** 3
**Originality:** 3
**Overall Recommendation:** 3
**Confidence:** 4

**Summary:**

This paper introduces PhoStream, a mobile streaming benchmark designed to evaluate the performance of omnimodal AI assistants in continuous real-world video streams. The benchmark unifies four scenarios: in-screen (e.g., mobile tutorials, app screen recordings) and out-of-screen (e.g., first-person perspective, vlogs). The dataset contains 5,572 open-ended question-and-answer pairs, constructed from an automatically generated pipeline and manually validated, with an average video length of 13.3 minutes. The paper designs an online inference pipeline with a 1-second update frequency, categorizing tasks into Instant, Backward, and Forward types. Experiments show that current large multimodal models exhibit severe "early response bias" in the "forward" task, responding prematurely before necessary cues appear, revealing a deficiency in timing judgment during streaming.

**Compliance With Llm Reviewing Policy:**

Affirmed.

**Final Justification:**

This paper introduces PhoStream, a timely benchmark for evaluating streaming multimodal assistants in long-form video, with strong practical relevance and improved ecological validity over prior work.

My main concerns focus on evaluation validity and analysis depth, and the rebuttal only partially addresses them:
- Long-video capability remains insufficiently validated. While the authors justify the 60-second sliding window and report a single longer-window result (10 min), they do not provide a systematic analysis across different window lengths (e.g., 30 / 60 / 90 / 120 seconds) to reveal performance trends. As a result, it is still unclear how much the benchmark truly evaluates long-range temporal reasoning versus being constrained by the chosen memory window.
- Audio effects are under-analyzed. The rebuttal suggests that audio may introduce earlier semantic cues that trigger premature responses, but this explanation remains high-level. There is no deeper investigation of the underlying mechanisms or supporting case studies. Several plausible factors could contribute, for example:
(1) Temporal misalignment: narration or sound cues may precede the corresponding visual evidence
(2) Modality dominance: models may overweight audio or video signals when uncertain
(3) Training bias to speech: models may be biased toward responding when confident semantic signals (e.g., speech) appear

Without targeted analysis or concrete examples, the findings remain observational rather than actionable. Given the central claim about “early response bias,” qualitative examples showing when and why models fail (with/without audio) would significantly strengthen the paper and help interpret the benchmark behavior.

I appreciate the hard work of the author, while the rebuttal improves clarity, it does not sufficiently address the core issues regarding long-context evaluation and modality interaction analysis. So I would like to keep my score.

**Key Questions For Authors:**

1. On evaluation rigidity.
The requirement to output “Silent” is counterintuitive. Among the cases labeled as “Early Response,” how many correspond to reasonable waiting-type responses (e.g., “I am monitoring and will respond when relevant”) that were penalized due to not matching the whitelist? If the evaluation criteria were relaxed, would the conclusions for forward tasks change?

2. On the 60-second sliding window.
The paper highlights long-video reasoning as a core challenge, yet the inference pipeline only accesses the most recent 60 seconds. How does this setup evaluate long-range temporal reasoning for backward tasks that depend on earlier context? Does this design introduce an artificial bottleneck?

3. On audio ablation analysis.
Table 4 shows that audio increases premature responses. Can the authors provide deeper analysis? For example, is this due to audio cues (e.g., narration) appearing earlier than corresponding visual events, thus triggering premature responses?

**Limitations:**

The current discussion of limitations is incomplete. While the paper briefly mentions potential privacy concerns in deployment, it does not address the methodological limitations of the evaluation pipeline itself. In particular, the 60-second sliding window and strict output constraints introduce significant evaluation bias and may distort the measured performance. A more explicit discussion of how these design choices affect the validity and interpretation of results would strengthen the paper.

**Strengths And Weaknesses:**

## Strength
- The paper advances evaluation of multimodal models from traditional offline video understanding to online streaming temporal reasoning, which is both timely and highly relevant to real-world applications. The unified mobile perspective that considers both on-screen and off-screen interactions is particularly meaningful from a practical deployment standpoint.
- Compared to prior streaming benchmarks that rely on multiple-choice questions or short clips (e.g., Proactive VideoQA with an average duration of 2.1 minutes), the use of longer videos (13.3 minutes on average) and open-ended QA better reflects realistic assistant interaction scenarios and increases ecological validity.


## Weakness
- The requirement for the model to output exactly the token “Silent” when evidence is insufficient is unintuitive and restrictive. If the model produces a reasonable intermediate response (e.g., a waiting or monitoring statement) that is not included in the limited whitelist (Listing A.1), it is penalized as an “Early Response” with zero score. This design can lead to false negatives and unfairly penalizes general-purpose models that exhibit reasonable interaction behavior but are not specifically tuned to this benchmark format.

- Although the paper emphasizes long videos (13.3 minutes), the online inference pipeline enforces a strict 60-second sliding window for memory efficiency. As a result, for backward reasoning tasks that require information beyond this window, failure is inevitable. This setup evaluates the limitations of the pipeline rather than the model’s actual long-context streaming reasoning ability.

- The paper observes that incorporating audio increases the rate of premature responses in forward tasks, which is an interesting finding. However, the analysis remains superficial and does not investigate underlying causes (e.g., potential temporal misalignment between audio narration and visual events). This limits the paper’s contribution to observation rather than providing actionable insights for improving streaming reasoning.

- The dataset is generated using Gemini 3 Pro and evaluated with LLM-as-a-Judge, which introduces potential self-preference bias. While a small-scale human evaluation (200 samples) is included, it is insufficient to fully validate the reliability of the full dataset (5,572 samples).

- Although the dataset construction process is relatively clear, the core online inference pipeline suffers from fundamental methodological issues. The combination of the 60-second sliding window and rigid output constraints undermines the validity of the benchmark, as the evaluation scores may not accurately reflect true streaming reasoning capability.

---

> ### Author Rebuttal · Authors · 2026-03-29
>
> **W1/Q1:** We understand your concern.
>
> (1) False-negative evaluation. To quantify possible FNs, we analyze all Early Response cases using the model output, video segment, question, and reference answer, and ask GPT-4o to label each response as Placeholder (e.g., “I am monitoring”), Correct, or Wrong.
>
> |Model|Correct(%)|Wrong(%)|Placeholder(%)|
> |-|-:|-:|-:|
> |Gemini-3-Pro|18.7|80.6|0.7|
> |Doubao-Seed-1.6|19.5|75.8|4.7|
> |Doubao-Seed-1.8|18.6|80.0|1.4|
> |Qwen3-Omni-30B-A3B|8.8|91.2|0.0|
> |Qwen3-VL-8B|8.2|91.7|0.1|
>
> Placeholder-like waiting responses are rare, up to 4.7% across models, while most ER are substantive responses, typically wrong guesses, rather than waiting text rejected by the whitelist. Thus, relaxing the exact-match criterion would affect a small fraction of cases and would not change our conclusion for Forward tasks.
>
> (2) Previous study. Similar "Silent"-style evaluation appears in prior benchmarks. For example, OmniMMI (CVPR 2025) computes proactive turn-taking accuracy as the percentage of instances with no response.
>
> **W2/Q2:** We agree that a 60s window constrains Backward questions whose key evidence lies earlier. However, PhoStream is designed to evaluate bounded-memory streaming assistance under a unified online protocol, not only long-context Backward reasoning.
>
> (1) Quantitative impact. Using GPT-4o, we estimate whether each Backward QA is answerable from the 60s context before the question timestamp. We find that 248 Backward samples require evidence from earlier than 60s, accounting for 22.1% of Backward QAs and 4.5% of all QAs. Thus, this limitation applies only to part of the Backward subset. Difficulty still lies in multimodal understanding and streaming reasoning.
>
> (2) Long videos matter. Long videos are important not only because they allow Backward questions to refer to earlier content, but also because they create richer streaming context over time. In online long videos, questions, cues, and answers are flexibly distributed over time, while video, audio, and dialogue continuously interleave and accumulate. This shows the value of our unified protocol for handling Backward, Instant, and Forward questions with a single procedure. It also makes PhoStream more challenging than prior short-clip open-ended benchmarks such as ProactiveVideoQA (2.1 min).
>
> (3) Benchmark scope. Streaming video is theoretically unbounded, and each model handles long-context streaming with its own memory-access strategy. Without a unified budget, differences would mainly reflect those mechanisms rather than streaming reasoning. We therefore fix a 60s sliding window for video and audio while keeping the full text dialogue history available for fair comparison.
>
> (4) Additional Backward experiment. Sliding window length is tunable. With a 10-min window, Gemini-3-Pro improves Backward from 82.2 to 85.6 and Qwen3-VL-8B from 71.5 to 73.0, confirming that longer context helps. However, we also observe failures on questions relying on recent evidence, suggesting that stronger streaming reasoning requires not only long-range access but also precise short-term grounding.
>
> **W3/Q3:** We provide more insights here.
>
> In Tab. 4, adding audio improves Instant and Backward performance for both Gemini 3 Pro and Qwen3-Omni, while slightly reducing Forward performance, increasing Early Response, and decreasing No Response. These results suggest that audio makes models more willing to answer.
>
> One reason is that audio may provide semantic cues earlier than the decisive visual evidence, but those cues are still incomplete at that point. For Forward questions that depend on future events or later visual confirmation, this can encourage an early hypothesis and a premature answer. This is consistent with the simultaneous increase in Early Response and decrease in No Response.
>
> **W4:** We would like to clarify two points:
>
> (1) The specific concern of judge-side self-preference does not apply because the LLM-Judge model is Qwen3-235B-A22B-Instruct rather than Gemini 3 Pro. We apologize for not stating this clearly; the judge-model configuration is included in the Supplementary Material code submitted with the paper.
>
> (2) Human evaluations in our work serve two different purposes. For question and answer correctness, human verification was conducted on the full dataset of 5,572 samples. For assessing LLM-as-a-Judge reliability, the human evaluation was conducted not on 200 QA samples, but on 200 videos (Line 433 of the paper) containing 1,810 QA pairs, approximately 1/3 of the dataset.
>
> **W5/L1:** We agree the 60s sliding window and the strict Silent rule may lead to limitations on scope and interpretation. These choices are deliberate parts of a bounded-memory streaming protocol. By design, PhoStream evaluates online streaming under bounded memory with a unified online inference protocol for fair comparison of streaming behavior across models. We thank you for your comments and will clarify these points in the revision.

---

> > ### Author Rebuttal · Reviewer_jQqU · 2026-04-08
> >
> > Please see final justification

---

> > > ### Author Response · Authors · 2026-04-08
> > >
> > > Thank you for the additional feedback. As this Rebuttal Acknowledgement was posted in the final hour of the discussion period, we do not have sufficient time to conduct careful new experiments within the rebuttal window. We therefore clarify the main points here and will include the requested analyses in the revision.
> > >
> > > (1) For long-video evaluation, the **average video length in PhoStream is 13.3 minutes (Table 1)**, so a 10-minute window already covers most cases in the dataset. We therefore believe the 10-minute result provides meaningful evidence beyond the default 60-second setting. A broader comparison across window lengths would be useful, and we will add it in the revision.
> > >
> > > (2) For the audio effect, we agree that deeper analysis would be valuable, but this is **not the main focus of the paper**. The core contribution is a scalable automated generative pipeline, a realistic online inference pipeline with LLM-as-a-Judge evaluation, and the identification of Early Response bias in streaming multimodal assistants.
> > >
> > > (3) We also note that the Appendix of paper **already provides qualitative evidence relevant to this concern**. In particular, Figures A.3 and A.4 include failure cases and modality-related examples on PhoStream. Figure A.3 shows representative Early Response cases from models such as Qwen3-Omni-30B-A3B and Qwen3-VL-30B-A3B, where the model responds before sufficient visual evidence has fully unfolded. Figure A.4 further shows a No Response case for the non-omni model Doubao-Seed-1.6/1.8, as well as examples where Gemini 3 Pro correctly leverage audio cues to interpret the scene. This supports our observation that audio can make models more willing to answer.
> > >
> > > We respectfully hope you will consider the main contribution of the paper, namely introducing a scalable and realistic benchmark and evaluation pipeline for streaming multimodal assistants, along with the identification of an important failure mode that merits further study.

---

### Decision · Program_Chairs · 2026-04-30

**Decision:**

Accept (regular)

**Comment:**

This paper proposes PhoStream, a mobile-centric streaming benchmark to unify on-screen and off-screen scenarios to evaluate video, audio, and temporal reasoning for MLLM. The studies reveal that current MLLM models perform well on Instant and Backward tasks, but drop significantly on Forward tasks largely due to early responses.

Initially, the reviewers raise a number of concerns, mainly on experiment setup, evaluation protocols and comparisons. In the rebuttal, the authors provide further clarifications and additional experimental results. Most reviewers acknowledge the the rebuttal and raised their initial score. One reviewer still have concerns on the verifications on long-video capability and audio effects, which I agree are valid concerns to be addressed, but is not likely to be done in the rebuttal period due to the time constraint. Having saying that, I agree the paper makes valid contributions on exploring the temporal reasoning gaps in MLLMs and the findings are valuable and appreciated by the reviewers. I highly encourage the authors to conduct additional studies in the final revision to address the remaining concerns on the verifications on long-video capability and audio effects.